# Budgeted Online Continual Learning by Adaptive Layer Freezing and Frequency-based Sampling

## Abstract

The majority of online continual learning (CL) places restrictions on the size of replay memory and a single-epoch training to ensure a prompt update of the model. However, the single-epoch training may imply a different amount of computations per CL algorithm, and additional storage for storing logit or model in addition to replay memory is largely ignored as a storage budget. Here, we used floating point operations (FLOPs) and total memory size in Byte as a metric for computational and memory budgets, respectively, to compare CL algorithms with the same total budget. Interestingly, we found that the new and advanced algorithms often perform worse than simple baselines under the same budget, implying that their value is less beneficial in real-world deployment. To improve the accuracy of online continual learners in the same budget, we propose an adaptive layer freezing and frequency-based memory retrieval for episodic memory usage for a storage- and computationally efficient online CL algorithm. The proposed adaptive layer freezing does not update the layers for less informative batches to reduce computational costs with a negligible loss of accuracy. The proposed memory retrieval balances the training usage count of samples in episodic memory with a negligible computational and memory cost. In extensive empirical validations using CIFAR-10/100, CLEAR-10, and ImageNet-1K datasets, we demonstrate that the proposed method outperforms the state-of-the-art in the same total budget.

## 1 Introduction

Most online continual learning (CL) research places restrictions such as the single training epoch to quickly update the model and the size of the replay memory that limits the number of streamed samples that can be stored (Koh et al., 2022; Wang et al., 2022a). While one-epoch training may give a rough sense of the computational constraint for each method, as each method requires a different amount of computations in an epoch, the actual computation budget to train the models differs per method. Additionally, several rehearsal-based CL methods require additional storage to store the past model and logit (*i.e.*, the unnormalized model output vector) (Buzzega et al., 2020; Koh et al., 2023; Zhou et al., 2023), which were not included in the replay memory size. Therefore, we attempt to rigorously compare CL methods with precise computational and memory constraints.

For a fair comparison of the methods in the same computational budget, we first consider the wall clock time of training as the metric. However, the wall clock time is highly dependent on hardware architectures, data I/O time, and the optimality of algorithm implementation (Gruber et al., 2022; Wintersperger et al., 2023; Prabhu et al., 2023). Thus, both the number of iterations and the training wall time may not be appropriate metrics for the computational cost of CL algorithms. In contrast, the Floating Point Operations (FLOPs) per sample is an exact computational budget regardless of the implementation details (Korthikanti et al., 2023). Following (Zhao et al., 2023; Ghunaim et al., 2023), we use training FLOPs as a metric for the computational budget.

For a fair comparison of the methods in the same memory budget, we need a total budget for various forms of extra storage including replay memory, logits, and model parameters. Following (Zhou et al., 2023), we convert all extra storage costs into Byte and sum them up to obtain the actual memory cost.

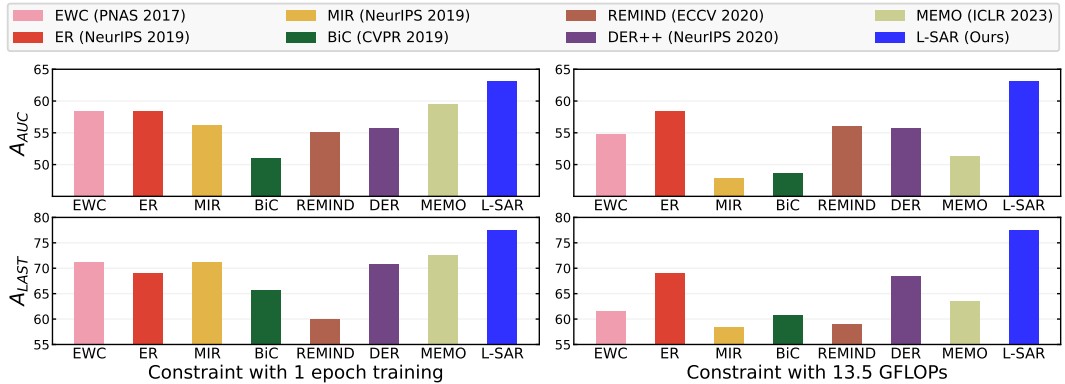

Figure 1: Comparison of CL methods with the same number of iterations and the same storage budget including the size of episodic memory and storage for past models for all methods (left) and our total-constrained CL considering training FLOPs per sample (right). (on CIFAR-10 Gaussian)

Taking into account the total memory and computational budget, we propose a computationally efficient online CL learning framework with negligible additional storage (0.02% of replay memory size in ImageNet), with a computation-aware layer freezing strategy. Specifically, we argue that, since not all layers are necessary to update for each data minibatch, if we find the appropriate layers to freeze for each mini-batch of data, we can reduce redundant training costs. We implement this idea by proposing 'adaptive layer freezing.' Since freezing earlier layers can save computational resources by reducing the computation of gradients in the backward pass (Hayes et al., 2020; Wu et al., 2020; He et al., 2021; Yuan et al., 2022), the frozen layers cannot learn any new information. Investigating the trade-off between computation and information due to freezing, we propose a novel method to choose the best layers to freeze by maximizing the Fisher Information(FI) gained by the model for each batch, given a fixed computation budget. Unlike previous freezing methods (Hayes et al., 2020; Lee et al., 2019; Yuan et al., 2022) that empirically select which layer to freeze, causing dependency on the dataset and the type of neural network, we consider the varying information of each batch, allowing us to determine the optimal layers to freeze at every forward pass.

While MIR (Aljundi et al., 2019a) and ASER (Shim et al., 2021) exhibit a substantial gain in accuracy, they demand high computational resources, as they require model inference on a large set of candidates. To obtain as much knowledge as possible on a limited total budget, we propose retrieving samples that the model has not learned much about from episodic memory. We utilize the frequency of recent use of each sample and the similarity of the gradient between classes, which are naturally obtained during training without requiring additional inference.

In our empirical validations, we compare the state-of-the-art CL algorithms in the literature under the same FLOPs of computations and the same Bytes of storage in Fig. 1. We observe that several high-performance CL methods are not competitive under fixed FLOPs and memory budget, interestingly, falling behind a simple Experience Replay (Rolnick et al., 2019). On the contrary, the proposed method outperforms them by a noticeable margin under the same computational and storage budget.

**Contributions.**   We summarize our contributions as follows:

- Proposing to rigorously measure computational and memory budgets of CL algorithms by using training FLOPs and total memory size in Bytes, to fairly compare different algorithms.
- Proposing a computationally efficient adaptive layer freezing that maximizes Fisher Information per computation.
- Proposing a memory retrieval strategy that promotes the retrieval of samples that the model has not learned much.
- Empirical analysis on the computational and memory costs of various CL algorithms, showing that many state-of-the-art CL methods are less beneficial under the same budget and showing that the proposed method outperforms them by a noticeable margin across multiple benchmarks.

## 2   RELATED WORK

**Online Continual Learning with Memory Budget.**   Replay-based online CL methods use episodic memory and consider the memory budget. Since we also consider the situation of using

episodic memory, we review them in detail as follows. The replay-based methods (Aljundi et al., 2019b; Prabhu et al., 2020; Bang et al., 2021; Koh et al., 2022; Wu et al., 2019) store part of the the past data stream in episodic memory to replay them in future learning.

Although there are simple sampling strategies such as random sampling (Guo et al., 2020) and reservoir sampling (Vitter, 1985), they are often insufficient to adapt to changing data distributions. Rather than simple methods, researchers have developed advanced sampling strategies considering factors such as uncertainty, diversity, and gradient (Lopez-Paz & Ranzato, 2017; Bang et al., 2021; Koh et al., 2022; Tiwari et al., 2022). However, these advanced methods often entail a high computational overhead, making them impractical for use in real-world applications. RM (Bang et al., 2021) requires a significant amount of computational cost to calculate the uncertainty for diversified sampling. Similarly, CLIB (Koh et al., 2022) involves an additional forward and backward stage to calculate the decrease in memory sample loss for each batch iteration.

Not only the memory management schemes, but researchers have also investigated the memory usage schemes, *i.e.*, sample retrieval strategies from the rehearsal buffers. In addition to random retrieval (Chaudhry et al., 2019), determining retrieval based on the degree of interference (Aljundi et al., 2019a) and the adversarial Shapley value (Shim et al., 2021) has been investigated. However, such methods require an inference of candidate samples, which leads to a nontrivial amount of computation in computing loss (Aljundi et al., 2019a) or the Shapely value (Shim et al., 2021).

**Computationally Efficient Learning using Layer Freezing.** Freezing layers have been investigated to reduce computational costs during training in joint training (*i.e.*, ordinary training scenario other than CL) (Brock et al., 2017; Goutam et al., 2020; Xiao et al., 2019). A common freezing approach (Wang et al., 2023; Li et al., 2022) includes determining whether to freeze a layer based on the reference model and representation similarity, such as CKA (Cortes et al., 2012) and SP loss (Tung & Mori, 2019). Additionally, EGERIA (Wang et al., 2023) unfreezes layers based on changes in the learning rate.

However, in CL, both online and offline, it is challenging to determine when to freeze a layer because metrics such as Euclidean distance and CKA cannot be used to compare the degree of convergence compared to the reference model (Mirzadeh et al., 2020). Additionally, continual learning involves a non-*i.i.d.* setup, where the data distribution continues to change (Criado et al., 2022). Therefore, in addition to changes in learning rate, it is important to consider the current data distribution when determining whether to freeze or unfreeze a layer in continual learning. Hayes et al. (2020) have explored freezing methods for continual learning. However, they use predefined freezing configurations such as the freezing backbone block 0 after task 1, while our freezing method adaptively freezes the layers using information per batch.

## 3 APPROACH

Training a neural network requires two passes of network traversal; forward and backward. To make learning efficient, we consider two strategies; (1) reducing the number of passes and (2) the computational cost of each pass. We address both aspects by proposing two components; an adaptive layer-freezing method and a new memory retrieval method. The layer freezing reduces the computational cost of each backward pass, which consumes twice the computations of the forward one. The memory retrieval method retrieves training batches that are insufficiently learned, so the model learns the same amount of knowledge in fewer iterations, reducing the number of training passes. Comprising the two proposals into a single framework, we call our method **Layer freezing and Similarity-Aware Retrieval (L-SAR)**. We illustrate our method in Fig. 2 and provide a pseudocode in Sec. A.2 in the appendix for the sake of space.

### 3.1 ADAPTIVE LAYER FREEZING FOR ONLINE CONTINUAL LEARNING

There have been several studies on the freezing of neural network layers in non-CL literature (Wang et al., 2023; Liu et al., 2021; He et al., 2021; Hinton et al., 2006). They suggest that freezing some layers can significantly reduce training computations with minimal impact on performance. These methods often rely on the convergence of each layer to determine which layers to freeze, since converged layers no longer require further training. However, in online CL, the model often

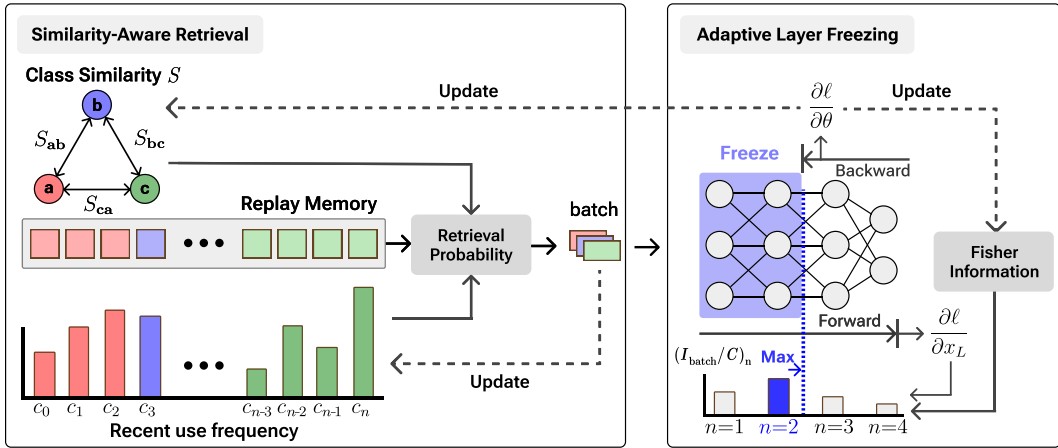

Figure 2: Overview of the proposed L-SAR. The colors in the 'Similarity-Aware Retrieval' box denote different classes. The dotted arrows denote copying the values, while the solid arrows denote the calculation of new values. 'Retrieval Probability' is calculated using class similarity $S$ and discounted use frequency $c_i$, where $c_i$ tracks the number of times sample $i$ has been used recently for training. A batch is sampled from memory by the 'Retrieval Probability' and $c_i$ is updated by retrieval results. After the forward pass of the model with the batch, we compute the freezing criterion $(I_{\text{batch}}/C)_n$ for each layer $n$ of the model, using Fisher Information and $\|\frac{\partial \ell}{\partial x_L}\|$. In the backward pass, layers 1 to $n_{\max} = \arg\max_n (I_{\text{batch}}/C)_n$ are frozen. Class similarity $S_{ij}$ and Fisher Information are updated using the gradient $\frac{\partial \ell}{\partial \theta}$ from the backward pass.

does not converge due to the limited training budget and the ever-evolving training data distribution, necessitating a new approach to decide when and which layers to freeze for incoming data.

**Selectively Freezing Layers by Maximum Fisher Information (FI).** For a computationally efficient freezing criterion, we propose freezing layers that learn little information per computation by measuring the amount of 'information' gained by each layer during training. Here, we define the information by the degree of certainty of the parameters with unknown true values (see Equation 1). With the information, we select the layers to freeze so that the model learns the maximum amount of information per computation. Since freezing reduces the computations for each iteration, we train a model with additional iterations in the same computational budget.

However, as a trade-off, freezing reduces the amount of information obtained per training iteration, since frozen layers do not gain information. To this end, to maximize the information ($I$) in the model while minimizing the computational cost ($C$), we propose to maximize the expected amount of information gained per computation ($I/C$). We factorize this by the amount of information gained per iteration ($I/\text{iter}$) and the number of iterations per computation (iter$/C$).

We first estimate ($I/\text{iter}$) when layers 1 to $n$ are frozen, which we denote as ($I/\text{iter}$)$_n$, for $n \in [1, L]$ where $L$ is the total number of layers. The amount of information obtained by layer $i$ is calculated by $\text{tr}(F(\theta_i))$, where $F(\theta_i)$ is the submatrix of Fisher Information Matrix $F(\theta)$ corresponding to layer $i$ and $\text{tr}(\cdot)$ is a trace operator. To efficiently calculate the information that each parameter acquires from the data, we use the diagonal component of the $F(\theta_i)$ since the diagonal components only require first-order derivatives rather than Hessian (Kirkpatrick et al., 2017; Soen & Sun, 2021) as:

$$(I/\text{iter})_n = \sum_{i=n+1}^{L} \text{tr}(F(\theta_i)), \quad \text{where} \quad F(\theta) = \mathbb{E}_{p_\theta(z)}\left[\left(\frac{\partial \ell}{\partial \theta}\right) \cdot \left(\frac{\partial \ell}{\partial \theta}\right)^{\mathsf{T}}\right], \quad (1)$$

where $\theta$ is the parameter of the model $p_\theta(\cdot)$, $z$ is the training data, and $\ell = \log p_\theta(z)$ is the loss function.

Now, we calculate (iter$/C$)$_n$ which refers to the number of iterations per computation when freezing up to layer $n \in [1, L]$. For notation brevity, we define Unit Computation (UC) as the total FLOPs required for a complete forward and backward pass of the model using a single batch. Formally, $\text{UC} = \sum_{i=1}^{L}(\text{BF})_i + (\text{FF})_i$, where $(\text{FF})_i$ and $(\text{BF})_i$ denote the forward FLOPs and the backward FLOPs of layer $i$, respectively. Without freezing, each iteration would cost 1 UC. In terms of UC,

we calculate $(\text{iter}/C)_n$ by the number of possible iterations given a computational budget of 1 UC, as:

$$(\text{iter}/C)_n = \frac{\text{UC}}{\text{UC} - \sum_{i=1}^{n} (\text{BF})_i}. \tag{2}$$

As the number of freezing layers increases (*i.e.*, $n$ increases), the possible iteration within the same computation increases.

Finally, combining Equation 1 and Equation 2, we can calculate expected amount of information gain per computation $I/C$ by a product of $I/\text{iter}$ and $\text{iter}/C$:

$$(I/C)_n = (I/\text{iter})_n \cdot (\text{iter}/C)_n = \sum_{i=n+1}^{L} \text{tr}(F(\theta_i)) \cdot \frac{\text{UC}}{\text{UC} - \sum_{i=1}^{n} (\text{BF})_i}. \tag{3}$$

Therefore, by freezing layer 1 to layer $n_{\max}$, where $n_{\max} = \arg\max_n (I/C)_n$, we can maximize the expected amount of information gained per computation during training.

**Batch-wise Version of $(I/C)$ for Online CL.** In online data stream, data distribution continuously shifts. Because of this, batches from past data distribution may contain less informative data, which makes it advantageous to freeze more layers, while batches from new distribution may contain much (*i.e.*, new) information, which makes it advantageous to freeze fewer layers. Thus, instead of determining $n_{\max}$ for the entire dataset, we adaptively freeze layers for each input batch by calculating $(I/C)_n$ per batch.

Since the FI is quadratically proportional to the magnitude of the gradient (see Equation 1), we estimate the information of each batch to be proportional to the squared gradient magnitude of the batch. To avoid full backpropagation, we only use the gradient of the last layer feature $x_L$ to estimate the gradient magnitude, following (Koh et al., 2023). For a detailed explanation of the gradient approximation using the last layer, please refer to Sec. A.1. Using these approximations, we obtain $(I_{\text{batch}}/C)_n$, the batch-wise version of $(I/C)_n$ as:

$$(I_{\text{batch}}/C)_n(z_{\text{t}}) = \frac{|\nabla_{x_L} \ell(z_{\text{t}})|^2}{\mathbb{E}_z \left[ |\nabla_{x_L} \ell(z)|^2 \right]} \cdot \sum_{i=n+1}^{L} \text{tr}(F(\theta_i)) + \left( \frac{\sum_{i=1}^{n} (\text{BF})_i}{\text{UC}} \right) \cdot \max_m (I/C)_m, \tag{4}$$

where $x_L$ represents the last layer features, and $I/C$ is defined in Equation 3. Please refer to Sec. A.1 for the detailed derivation of Equation 4. Here, we compute the average gradient magnitude $\mathbb{E}_z \left[ |\nabla_{x_L} \ell(z)|^2 \right]$ of the last layer and the trace of Fisher Information $\text{tr}(F(\theta_i)) = \mathbb{E}_{p_\theta(z)} \left[ \text{tr} \left( \left( \frac{\partial \ell}{\partial \theta_i} \right) \cdot \left( \frac{\partial \ell}{\partial \theta_i} \right)^{\mathsf{T}} \right) \right] = \mathbb{E}_z \left[ \sum (\nabla_{\theta_i} l(z))^2 \right]$ for all layers $i \in [1, L]$. Since calculating the expected values (using all samples) in every learning iteration is computationally expensive, we estimate them by exponential moving average (EMA) of the estimated expectations computed by the mini-batch of the past iterations. However, the EMA estimate of $\text{tr}(F(\theta_i))$ requires a gradient calculation for all layers, so it cannot be used with freezing, which stops gradient computations. Since the estimation of $\text{tr}(F(\theta_i))$ and freezing cannot be performed at the same time, at each $m$ iteration, we train (*i.e.*, unfreeze) all layers to update the estimate of $\text{tr}(F(\theta_i))$ for all $i$. For the other $m - 1$ iterations, we do not update $\text{tr}(F(\theta_i))$ and freeze the model based on the values of $I_{\text{batch}}/C$, using the previously estimated value of $\text{tr}(F(\theta_i))$.

In summary, we maximize the information learned given a fixed computational budget by freezing layer 1 to $n$ that maximizes $(I_{\text{batch}}/C)_n(z_{\text{t}})$ for batch $z_t$. We argue that it allows us to dynamically adjust the degree of freezing based on the learning capacity of the layers and the batch informativeness.

## 3.2 SIMILARITY-AWARE RETRIEVAL BASED ON 'USE-FREQUENCY'

In rehearsal-based CL methods, sample retrieval strategies such as MIR (Aljundi et al., 2019a) and ASER (Shim et al., 2021) do not consider the computational cost. Despite the computational costs, some strategies perform worse than random retrieval (Fig. 1). Here, we propose a computationally efficient sample retrieval strategy.

In online CL, new data continuously streams in, and old data remains in memory, causing an imbalance in 'the number of times each sample is used for training', which we call '*use-frequency*.' We

argue that samples with high use-frequency have been sufficiently learned by the model, and additional training with them provides a marginal knowledge gain while incurring computational costs. In contrast, samples with low use-frequency are likely to contain the knowledge that the model learns insufficiently. Therefore, we give these samples a higher probability of being retrieved.

Additionally, if a sample was frequently used in the past but less frequently in recent iterations, its knowledge may have been forgotten, despite a high use-frequency. Inspired by the exponential decaying model of forgetting (Shin & Lee, 2020; Mahto et al., 2020; Chien et al., 2021), we propose a decay factor in the use frequency at each iteration, denoted as $0 < r < 1$, resulting in the '*discounted-use-frequency.*'

**Effective Use Frequency** ($\hat{c}_i$). However, the model can learn knowledge about the sample by training other samples that are similar in the same class. This *effectively* increases the use-frequency for that sample. On the contrary, the model may lose knowledge about the sample when training on samples from other classes, effectively decreasing the use-frequency.

Inspired by Dhaliwal & Shintre (2018); Du et al. (2018), we assume that samples with similar gradients have similar information and effectively increase the use frequency, while those with opposite gradients would effectively decrease the use-frequency. To account for this, we define *'effective-use-frequency'* by adding the other samples' use-frequency multiplied by the cosine similarity between gradients. However, since tracking the gradient similarities between all sample pairs requires excessive memory ($\sim 10^{12}$ for ImageNet) and computation, we use class-wise similarities, which is the expected gradient similarity between samples from two classes. Formally, we define the class similarity $\mathcal{S}_{y_1,y_2}$ for classes $y_1$ and $y_2$ as:

$$\mathcal{S}_{y_1,y_2} = \mathbb{E}_{z_1 \in D_{y_1}, z_2 \in D_{y_2}} \left[\cos(\nabla_\theta l(z_1), \nabla_\theta l(z_2))\right] \tag{5}$$

where $D_{y_i}$ is the training data for class $y_i$. Using class-wise similarities, we calculate the effective-use-frequency $\hat{c}_i$ for the sample $i$ as

$$\hat{c}_i = c_i + \sum_{y \in \mathcal{Y}} \mathcal{S}_{y,y_i} \cdot C_y, \tag{6}$$

where $c_i$ is the discounted-use-frequency for sample $i$, $\mathcal{S}_{y,y_i}$ is a class similarity between class $y$ and $y_i$, and $C_y$ is the sum of the discounted-use-frequencies for all samples of class $y$.

Calculating the expected value in Equation 5 from scratch requires a gradient calculation for all samples in class $y_1$ and $y_2$, which is computationally expensive. As a computationally efficient alternative, we use the EMA to update the previous estimate rather than to evaluate the expectation from scratch. Note that we reuse the gradients obtained during training to calculate similarity and update the EMA estimate of $\mathcal{S}_{y_i,y_j}$. Specifically, we calculate the cosine similarity of the gradients between all pairs of samples in the training batch and update the EMA estimate of $S_{y_iy_j}$ using it, where $y_i$ and $y_j$ are labels for each pair. We only use the layers that are not frozen for this calculation.

To further reduce the computational cost of calculating similarity, we use only $0.05\%$ of the model parameters for the calculation of similarity, since the gradient distribution of the subset of randomly selected weights is similar to the gradient of the entire weight set (Li et al., 2022).

Finally, we obtain the retrieval probabilities $p_i$ for $i$-th sample by the softmax of $-\hat{c}_i/T$, where $T$ is a temperature hyper-parameter, as

$$p_i = \frac{e^{-\hat{c}_i/T}}{\sum_{j=1}^{|\mathcal{M}|} e^{-\hat{c}_j/T}}. \tag{7}$$

Samples with a low $\hat{c}_i$ have a higher chance of being retrieved. This allows the model to prefer learning relatively insufficiently trained samples to sufficiently trained ones, thus accelerating the training. Note that this retrieval strategy uses information that is naturally obtained during training, such as use-frequency and gradients, imposing negligible additional computations.

## 4 EXPERIMENTS

### 4.1 EXPERIMENTAL SETUP

For empirical validation, we adopt the total budget for memory and computation. For the memory budget, we use Bytes (Zhou et al., 2023), which considers memory costs not only for the samples

in episodic memory but also for additional model parameters used in regularization or distillation. For the computational budget, we use FLOPs in the training phase. For dataset, we use CIFAR-10, CIFAR-100, ImageNet, and CLEAR-10. We evaluate the methods in conventional disjoint CL task setup and a newly proposed Gaussian task setup for boundary-free continuous data stream (Shanahan et al., 2021; Wang et al., 2022b; Koh et al., 2023). For all experiments, we averaged 3 different random seeds, except ImageNet due to computational cost (Bang et al., 2021; Koh et al., 2023). We conducted a Welch's $t$-test with a significance level of 0.05. If the highest performance in each column is statistically significant, it is highlighted in bold. Otherwise, it is underlined.

**Metrics.** We report the last accuracy $A_{\text{last}}$ and the area under the curve of accuracy $A_{\text{AUC}}$ (Koh et al., 2022). The $A_{\text{last}}$ measures the accuracy at the end of CL. The $A_{\text{AUC}}$ measures the average accuracy throughout the training course. To calculate $A_{\text{AUC}}$, we use evaluation period as 100 samples for CIFAR-10, CIFAR-100 and CLEAR-10, 8000 samples for ImageNet. For each evaluation, we use the entire test set for the class seen so far as the evaluation set. We argue that $A_{\text{AUC}}$ is a suitable metric to measure prompt learning of new knowledge.

**Baselines.** We compare our method to Experience Replay (ER) (Rolnick et al., 2019), Dark Experience Replay (DER++) (Buzzega et al., 2020), Maximally Interfered Retrieval (MIR) (Aljundi et al., 2019a), Memory-efficient Expandable Model (MEMO) (Zhou et al., 2023), REMIND (Hayes et al., 2020), Elastic Weight Consolidation (EWC) (Kirkpatrick et al., 2017) and Bias Correction (BiC) (Wu et al., 2019).

We describe the details of the implementation in Sec. A.4 in the Appendix for the sake of space.

## 4.2 RESULTS

We evaluate CL methods including L-SAR, with strictly restricted computation and memory budgets as specified in Sec. 4.1. Note that, unlike other methods, which could be adjusted to have the same FLOPs/sample by controlling the number of iterations/sample, L-SAR adaptively reduces FLOPs through adaptive layer freezing. Therefore, we set the (iteration/sample) of L-SAR to 1, which is the same as the (iteration/sample) of ER, which costs the least computation among the baselines.

**Various Computational Budget under the Same Memory Budget.** First, we compare CL methods under fixed memory budgets and various computational budgets in Fig. 3. We observe that L-SAR significantly outperforms other methods in all datasets and both Gaussian and disjoint task setups except ImageNet-Disjoint, especially under a low computational budget. It shows that our layer freezing and similarity-aware retrieval generally effectively reduce the computational cost, especially when the computational budget is tight. In ImageNet-Disjoint setup, some other methods show comparable performance with L-SAR. In that setup, since a large batch size of 256 is used, the data distribution in each batch does not change much, and so does the amount of total information in each batch, leading to less gain by our freezing method, which considers the information of each batch. Note that the disjoint CL setup is argued as less realistic scenarios (Prabhu et al., 2020; Bang et al., 2021; Koh et al., 2022), but we use it since many methods are proposed for that.

Additionally, we observe an increase in the FLOPs saved by L-SAR through freezing, particularly pronounced at higher computational budgets. As the model is trained for more iterations, the amount of information the model gains from the training data decreases. Thus, our adaptive layer freezing adaptively adjusts the freezing criterion to freeze more layers, leading to lower FLOPs, thus the line stops at the earlier GFLOPs value than the compared methods.

**Various Memory Budget under the Same Computational Budget.** We now fix the computational budget and test various memory budgets for CL methods, and summarize the results in Tab. 1 for CIFAR-100. L-SAR again outperforms other methods by a significant margin in all datasets, implying that L-SAR is suitable for both large and small memory budgets. Since L-SAR uses minimal additional memory in addition to episodic memory and utilizes episodic memory effectively by similarity-aware retrieval, L-SAR consistently outperforms other methods in various memory sizes. Please refer to Sec. A.8 for results with various memory budgetss in CIFAR-10.

We investigate CL methods in domain incremental setup with fixed computational and memory budget using the CLEAR-10 dataset, in Tab. 2. Unlike the class incremental, where new classes are introduced to the model, the domain incremental introduces new samples that are in different domains, while the classes are maintained the same. As shown in the table, L-SAR also outperforms the state-of-the-art in domain incremental setups.

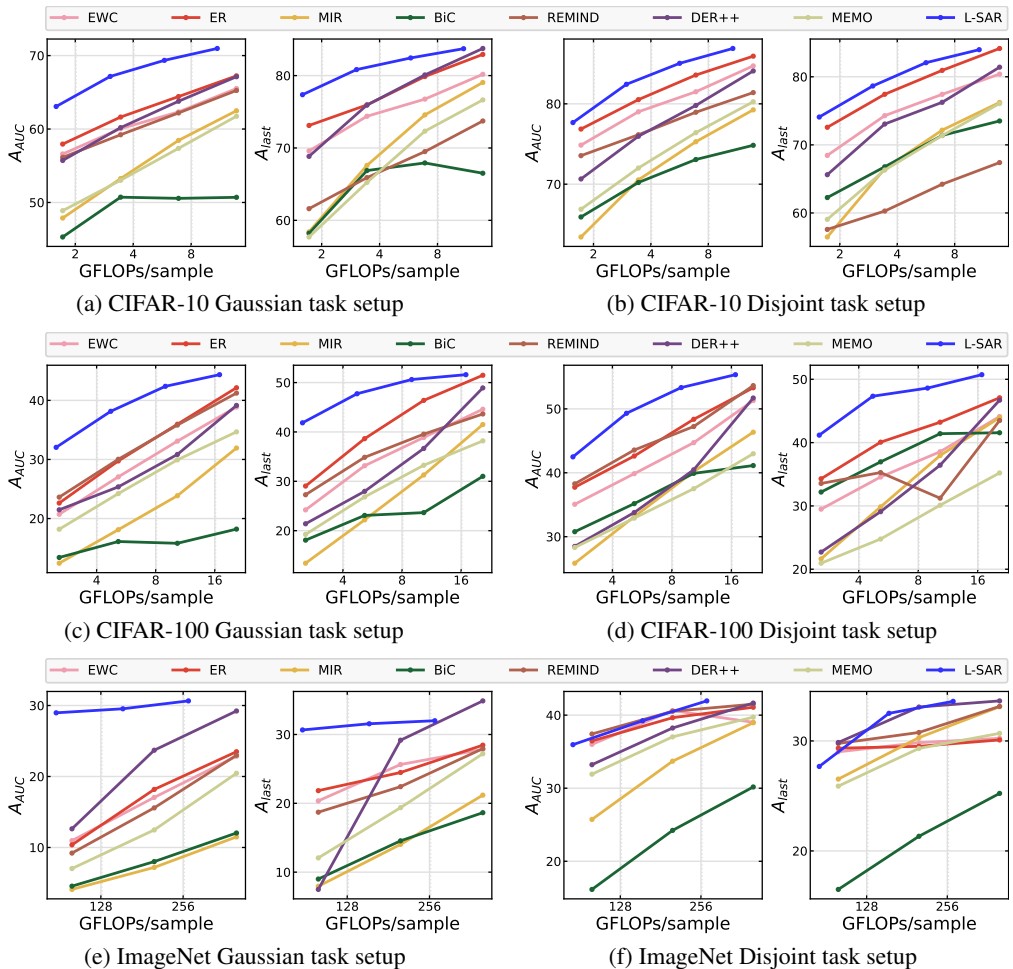

Figure 3: Accuracy on Gaussian and Disjoint CL setup in CIFAR-10, CIFAR-100, and ImageNet for a wide range of FLOPs per sample. L-SAR outperforms all CL methods compared.

| Methods | Memory Size | | | | | |
| | 7.6 MB | | 13.44 MB | | 25.12 MB | |
| | $A_{\text{AUC}} \uparrow$ | $A_{\text{last}} \uparrow$ | $A_{\text{AUC}} \uparrow$ | $A_{\text{last}} \uparrow$ | $A_{\text{AUC}} \uparrow$ | $A_{\text{last}} \uparrow$ |
|---|---|---|---|---|---|---|
| ER (Rolnick et al., 2019) | 22.56±1.61 | 27.52±1.90 | 22.95±1.62 | 29.93±0.82 | 22.62±2.15 | 29.04±2.58 |
| REMIND (Hayes et al., 2020) | 22.86±1.32 | 24.91±1.40 | 23.60±1.47 | 26.60±1.76 | 23.62±1.11 | 27.28±0.61 |
| DER++ (Buzzega et al., 2020) | 21.56±0.87 | 21.07±0.41 | 21.66±1.07 | 21.46±1.32 | 21.48±1.03 | 21.40±1.65 |
| ER-MIR (Aljundi et al., 2019a) | 12.13±2.39 | 13.13±3.09 | 12.91±1.83 | 13.72±2.26 | 12.44±2.28 | 13.41±2.77 |
| EWC (Kirkpatrick et al., 2017) | 19.27±1.37 | 19.75±1.50 | 20.67±2.09 | 24.34±2.43 | 20.72±2.65 | 24.21±2.28 |
| BiC (Wu et al., 2019) | 21.57±0.64 | 27.93±0.58 | 29.79±1.76 | 28.23±2.92 | 16.11±1.24 | 23.09±0.52 |
| MEMO (Zhou et al., 2023) | 26.61±0.37 | 17.46±1.26 | 29.65±0.51 | 30.56±0.61 | 29.93±0.61 | 33.25±0.62 |
| L-SAR (Ours) | **30.57±0.62** | **34.00±0.51** | 31.72±0.51 | **38.46±0.92** | **32.05±0.59** | **41.85±0.59** |

Table 1: Accuracy for different memory sizes for Gaussian data Stream in CIFAR-100. The computational budget is fixed as 128.95 TFLOPs.

We believe this is because our retrieval method balances the use-frequency of samples in different domains so that the model learns more on relatively less-learned domains, allowing fast adaptation to new domains. It is more prominent in the results that L-SAR outperforms other methods by a larger gain in $A_{\text{AUC}}$ than in $A_{\text{last}}$, where $A_{\text{AUC}}$ measures the accuracy of all time. Note that L-SAR also saves a significant amount of FLOPs thanks to adaptive layer freezing.

## 4.3 ABLATION STUDY

We now ablate the model to investigate the benefit of each of the proposed components by using CIFAR-10 and CIFAR-100 in the Gaussian task setup and summarize the results in Table 3. See

| Metric | EWC | ER | ER-MIR | BiC | REMIND | DER++ | MEMO | L-SAR (Ours) |
|---|---|---|---|---|---|---|---|---|
| $A_{\text{AUC}}$ ↑ | 63.40±0.17 | 64.61±0.15 | 59.02±0.31 | 51.20±0.46 | 63.86±0.37 | 61.55±0.51 | 58.24±0.65 | **68.47±0.17** |
| $A_{\text{last}}$ ↑ | 73.85±1.29 | 75.58±0.70 | 70.40±0.69 | 66.57±1.61 | 75.05±0.23 | 73.70±0.46 | 67.64±1.92 | 76.57±0.58 |
| TFLOPs ↓ | | | | 2,640.37 | | | | **2,104.20** |

Table 2: Accuracy of various CL methods in domain-IL setup with CLEAR-10 dataset. Our L-SAR outperforms other methods with a smaller computational budget and same storage budget.

| Methods | CIFAR-10 | | | CIFAR-100 | | |
|---|---|---|---|---|---|---|
| | $A_{\text{AUC}}$ ↑ | $A_{last}$ ↑ | TFLOPs ↓ | $A_{\text{AUC}}$ ↑ | $A_{last}$ ↑ | TFLOPs ↓ |
| Vanilla | 60.76±0.11 | 70.08±0.97 | 163.74 | 31.97±0.89 | 37.80±1.30 | 245.91 |
| + Freezing | 60.38±0.54 | 69.04±0.83 | **142.23** | 31.77±0.60 | 38.03±0.35 | **217.40** |
| + Retrieval | 64.60±0.83 | 72.43±0.38 | 171.94 | 37.60±0.40 | 42.69±0.18 | 257.97 |
| + Retrieval & Freezing (L-SAR) | 64.38±0.32 | 72.57±0.79 | 146.80 | 37.20±0.73 | 42.55±0.79 | 221.49 |

Table 3: Benefits of the proposed components of our method in CIFAR-10 and CIFAR-100 for Gaussian task setup. The memory budget is 7.6MB for CIFAR-10 and 13.44MB for CIFAR-100. CIFAR-10 We train for 1 iter per sample for CIFAR-10 and 1.5 iter per sample for CIFAR-100.

Sec. A.7 in Appendix for ablation in the disjoint setup. For a comparison between adaptive layer freezing and naive layer freezing methods, please refer to Appendix Sec. A.6.

'Vanilla' is a simple replay-based method that trains on randomly retrieved batches from a balanced reservoir memory. As shown in the table, our similarity-aware retrieval based on use-frequency increases the performance of the baseline in the same number of iterations. This shows that our retrieval method increases the amount of knowledge learned per iteration, as we claim in Sec. 3.

While computational cost also increases, its increase is modest compared to other retrieval methods such as MIR (Aljundi et al., 2019a) or ASER (Shim et al., 2021) which require $2 \sim 3\times$ more computations. Also, we observe that the adaptive layer freezing method saves a significant amount of FLOPs while preserving accuracy. This shows that our freezing method effectively reduces the computational cost of each iteration as claimed in Sec. 3, with a negligible drop in performance. Summing up the effect of the two components, our method outperforms the baseline while using fewer FLOPs than the baseline, each by a noticeable margin. We show the effect of freezing on accuracy and FLOPs as the training progresses, in Fig. 4.

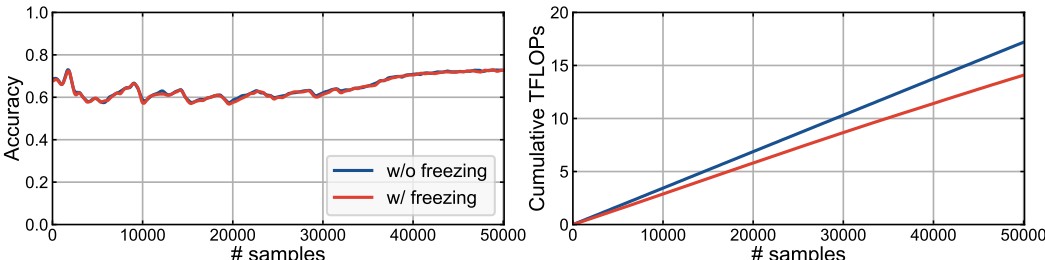

Figure 4: Accuracy and computational cost of the adaptive layer freezing in L-SAR. Training for 1 iteration per sample in CIFAR-10 Gaussian task setup.

## 5 CONCLUSION

We address the challenge of achieving high performance on both old and new data with minimal computational cost and a limited storage budget in online CL. While CL with fixed episodic memory size has been extensively studied, we have investigated the total storage budget required for the online CL as well as the computational budget for developing practically useful online CL methods. To this end, we proposed L-SAR, a computation-efficient CL method comprising two components: similarity-aware frequency-based retrieval and adaptive layer freezing. Our empirical validations show that several high-performing CL methods are not competitive under a fixed computational budget, falling behind a simple baseline of training on randomly retrieved batches from memory.

**Limitations and Future Work.** While our method only requires negligible additional memory other than episodic memory, it does not actively optimize for the memory efficiency of CL algorithms. It is interesting to explore a method to use the limited storage budget more efficiently, *e.g.*, storing quantized versions of models and exemplars.

ETHICS STATEMENT

We propose a better learning scheme for online continual learning for realistic learning scenarios. While the authors do not explicitly aim for this, the increasing adoption of deep learning models in real-world contexts with streaming data could potentially raise concerns such as inadvertently introducing biases or discrimination. We note that we are committed to implementing all feasible precautions to avert such consequences, as they are unequivocally contrary to our intentions.

REPRODUCIBILITY STATEMENT

We take reproducibility in deep learning very seriously and highlight some of the contents of the manuscript that might help to reproduce our work. We will definitely release our implementation of the proposed method in Sec. 3, the data splits and the baselines used in our experiments in Sec. 4

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

# A APPENDIX

## A.1 DERIVATION OF BATCH-WISE INFORMATION PER COMPUTATION

Computing the exact FI for each batch requires exact gradients for all layers in each batch, and it requires a full backward pass, making efforts to save the backward cost of frozen layers futile.

To address this problem, we propose to estimate FI for each batch from the previously calculated FI for the entire data distribution and the magnitude of the gradient of the batch, using the quadratic scaling relationship between the FI and the magnitude of the gradient, as shown in Equation 1. To obtain the magnitude of the gradient without full backpropagation, we calculate it solely using the last layer's feature representation, since the gradient magnitudes for the preceding layers would scale with the last layer by the chain rule (Koh et al., 2023).

This allows the batch-wise version of $(I/\text{iter})$ by multiplying the ratio of the current batch $(z_t)$'s gradient magnitude $(|\nabla_{x_L}\ell(z_t)|^2)$ and the average gradient magnitude $(\mathbb{E}_z\left[|\nabla_{x_L}\ell(z)|^2\right])$ for entire training data $z$, to obtain $(I_{\text{batch}}/\text{iter})$ as:

$$(I_{\text{batch}}/\text{iter})_n(z_t) = \frac{|\nabla_{x_L}\ell(z_t)|^2}{\mathbb{E}_z\left[|\nabla_{x_L}\ell(z)|^2\right]} \cdot (I/\text{iter})_n = \frac{|\nabla_{x_L}\ell(z_t)|^2}{\mathbb{E}_z\left[|\nabla_{x_L}\ell(z)|^2\right]} \cdot \sum_{i=n+1}^{L}\text{tr}(F(\theta_i)), \quad (8)$$

where $x_L$ is the last layer features. Using $(I_{\text{batch}}/\text{iter})$, we define the batch-wise version of $(I/C)$ (Equation 3) to measure the usefulness of the current batch $z_t$, $(I_{\text{batch}}/C)$ as:

$$(I_{\text{batch}}/C)_n(z_t) = (I_{\text{batch}}/\text{iter})_n(z_t) \cdot 1 + \frac{(\text{iter}/C)_n - 1}{(\text{iter}/C)_n} \cdot \max_m (I/C)_m \quad (9)$$

$$= \frac{|\nabla_{x_L}\ell(z_t)|^2}{\mathbb{E}_z\left[|\nabla_{x_L}\ell(z)|^2\right]} \cdot \sum_{i=n+1}^{L}\text{tr}(F(\theta_i)) + \left(\frac{\sum_{i=1}^{n}(\text{BF})_i}{\text{UC}}\right) \cdot \max_m (I/C)_m, \quad (10)$$

where $x_L$ represents the last layer features, and $I/C$ is defined in Equation 3. The first term in the equation represents the FI obtained from a single iteration with the current batch, which is $(I_{\text{batch}}/\text{iter})$. The second term is the expected FI gain by using the computation saved by freezing, obtained as a product of the amount of saved computation and the maximum value of $I/C$.

## A.2 DETAILED ALGORITHM OF L-SAR

Algorithm 1 provides a comprehensive pseudocode for the L-SAR method. L-SAR has two components: similarity-aware retrieval and adaptive layer freezing. In the algorithm box, lines 3, 6-13, and 25-26 describe the similarity-aware retrieval method, and lines 15-24 describe the adaptive layer freezing method.

## A.3 DETAILS ABOUT THE MEMORY BUDGET IN TOTAL-CONSTRAINED CL

The memory budget is allocated to episodic memory, model parameters, and additional memory costs specific to each CL algorithm, such as classwise similarities and logits. In this section, $\mathcal{B}$ denotes the additional memory budget, $S(|\mathcal{B}|)$ denotes the size of the additional memory budget (in MB), $\mathcal{E}$ denotes episodic memory, and $|\mathcal{E}|$ represents the number of stored instances in $\mathcal{E}$.

In our total-constrained setup, the memory budget is restricted to the cost of storing 7.6MB, 13.44MB and 25.12 MB in CIFAR-10/100. Since storing the ResNet-32 model parameters requires memory cost equivalent to saving 603 instances of CIFAR-100 images (463,504 floats × 4 bytes/float ÷ (3 × 32 × 32) bytes/image ≈ 603 instances), for methods that store the model for distillation or regularization, we subtract the memory cost of the model parameters from the episodic memory size (Zhou et al., 2023). In ImageNet and CLEAR-10, we use the ResNet-18 model and apply the same policy of subtracting model parameters and logits from the memory budget as mentioned above.

ER does not require additional memory beyond episodic memory. Similarly, MIR (Aljundi et al., 2019a) and ASER (Shim et al., 2021) do not require additional memory despite being computationally heavy.

---

**Algorithm 1** Layer freezing and Similarity-Aware Retrieval (L-SAR)

---

1: **Input** model $f_\theta$, Layer parameters $\theta_l$, Training data stream $\mathcal{D}$, Batch size $B$, Learning rate $\mu$, EMA ratio $\alpha$, Frequency scale $k$, Retrieval temperature $T$, Number of layers $L$, Total FLOPs (TF), Backward FLOPs per layer $(\text{BF})_l$

2: **Initialize** Episodic memory $\mathcal{M} \leftarrow \{\}$, Sample frequency $c_i \leftarrow 0$, Class frequency $C_y \leftarrow 0$, Class Similarity $S_{y_1 y_2} \leftarrow 0$, Layer Fisher trace $(trF)_l \leftarrow 0$, Expected gradient norm $|\bar{g}_{x'_L}| \leftarrow 0$

3: $\theta_S = \text{RandomSubset}(\theta, 0.0005)$
  $\triangleright$ Random subset of $\theta$ containing $0.05\%$ of the parameters, for updating class similarity $S$

4: **for** $(x_t, y_t) \in \mathcal{D}$ **do** $\hspace{2cm} \triangleright$ samples from data stream

5: $\quad$ Update $\mathcal{M} \leftarrow \text{GreedyBalancingSampler}\left(\mathcal{M} \cup (x_t, y_t)\right)$
  $\triangleright$ Memory update with Greedy Balancing Sampler

6: $\quad \hat{c}_i = c_i + \sum_{y \in \mathcal{Y}} S_{yy_i} C_y \quad \forall (x_i, y_i) \in \mathcal{M}$
  $\triangleright$ Calculate effective-use-frequency by Eq. (6)

7: $\quad \mathcal{I} = \text{RandomChoice}(|\mathcal{M}|, B, \text{softmax}(e^{-\hat{c}_i/T})) \quad \triangleright$ Sample batch indices from memory

8: $\quad r = \frac{B}{k|\mathcal{M}|} \hspace{5cm} \triangleright$ Calculate decay rate

9: $\quad$ Update $c_i \leftarrow (1-r)c_i \quad \forall (x_i, y_i) \in \mathcal{M} \hspace{1cm} \triangleright$ Decay the sample frequencies

10: $\quad$ Update $C_y \leftarrow (1-r)C_y \quad \forall y \in \mathcal{Y} \hspace{1.5cm} \triangleright$ Decay the class frequencies

11: $\quad$ Update $c_i \leftarrow c_i + 1 \quad \forall i \in \mathcal{I} \hspace{1cm} \triangleright$ Increase sample frequency for selected samples

12: $\quad$ Update $C_{y_i} \leftarrow C_{y_i} + 1 \quad \forall i \in \mathcal{I} \hspace{1cm} \triangleright$ Increase class frequency for selected samples

13: $\quad z_t = \{(x_i, y_i) \quad \forall i \in \mathcal{I}\} \hspace{3cm} \triangleright$ Obtain training batch $z_t$

14: $\quad \mathcal{L}(z_t) = \sum_{(x,y) \in z_t} \text{CrossEntropy}(f_\theta(x), y) \hspace{1cm} \triangleright$ Calculate loss

15: $\quad g_{x'_L}(z_t) = \nabla_{x'_L} \mathcal{L}(z_t) \hspace{3cm} \triangleright$ Obtain gradient for last feature $x'_L$

16: $\quad$ **if** $t\%4 = 0$ **then**

17: $\quad\quad$ **Update** $(trF)_l \leftarrow (1-\alpha)(trF)_l + \alpha \sum (\nabla_{\theta_l} \mathcal{L}(z_t))^2 \quad \forall l \in 1, \dots L$
  $\triangleright$ Update Fisher every 4 batches

18: $\quad\quad n^* = 0 \hspace{5cm} \triangleright$ No freezing When Fisher update

19: $\quad$ **else**

20: $\quad\quad (I/C)_n = \frac{(\text{TF})}{(\text{TF}) - \sum_{l=1}^{n}(\text{BF})_l} \sum_{l=n+1}^{L} (trF)_l \quad \forall n \in 1, \dots, L \hspace{0.3cm} \triangleright$ Compute $(I/C)$ by Eq. (3)

21: $\quad\quad (I_{\text{batch}}/C)_n(z_t) = \frac{|g_{x'_L}(z_t)|^2}{|\bar{g}_{x'_L}|^2} \cdot \sum_{l=n+1}^{L} (trF)_l + \frac{\sum_{l=1}^{n}(\text{BF})_l}{\text{TF}} \cdot \max_m (I/C)_m$
  $\forall n \in 1, \dots, L \hspace{4cm} \triangleright$ Compute $(I_{\text{batch}}/C)$ by Eq. (4)

22: $\quad\quad n^* = \text{argmax}_n (I_{\text{batch}}/C)_n(z_t) \hspace{1.5cm} \triangleright$ Determine optimal freezing

23: $\quad$ **end if**

24: $\quad$ **Update** $|\bar{g}_{x'_L}| \leftarrow (1-\alpha) \cdot |\bar{g}_{x'_L}| + \alpha \cdot |g_{x'_L}(z_t)| \triangleright$ Update expected gradient norm for last feature

25: $\quad \theta_{S,n^*} = \theta_S \cap \theta_{(n^*+1,\dots,L)} \hspace{2cm} \triangleright$ Use only unfrozen parameters for updating similarity

26: $\quad$ **Update** $S_{y_i y_j} \leftarrow (1-\alpha)S_{y_i y_j} + \alpha \cdot \text{CosineSimilarity}\left(\nabla_{\theta_{S,n^*}}^{(i)} \mathcal{L}(z_t), \nabla_{\theta_{S,n^*}}^{(j)} \mathcal{L}(z_t)\right)$
  $\forall (x_i, y_i), (x_j, y_j) \in z_t, i \neq j \hspace{1.5cm} \triangleright$ Update class similarity using sample-wise gradients

27: $\quad$ **Update** $\theta_{(n^*+1,\dots,L)} \leftarrow \theta_{(n^*+1,\dots,L)} - \mu \cdot \nabla_{\theta_{(n^*+1,\dots,L)}} \mathcal{L}(z_t)$
  $\triangleright$ Update the model except frozen layers

28: **end for**

29: **Output** $f_\theta$

---

On the contrary, EWC (Kirkpatrick et al., 2017) requires storing the previous model parameters and the parameter-wise Fisher Information(FI) for all parameters. Therefore, we subtract the memory cost of storing two models from the episodic memory size. Similarly, BiC Wu et al. (2019) also stores the previous model for distillation, $|\mathcal{E}|$ was reduced as much as the size of the model. For example, with a total memory budget of 7.6MB and a ResNet-32 model type in CIFAR-100, ER can store up to 2000 instances in $\mathcal{E}$, while EWC is limited to storing only 794(=2000 - 2×603) instances. Similarly, BiC can store only 1397(=2000 - 603) instances in $\mathcal{E}$.

Some methods incur additional memory costs other than episodic memory or model parameters. We handle such costs in a similar way by reducing the episodic memory size by the number of samples equivalent to the additional memory cost. For example, DER (Buzzega et al., 2020) uses the previous logits of the samples for distillation, so we subtract the cost of storing the logits from

| Methods | $\mathcal{B}$ Type | $S(|\mathcal{B}|)$ | $|\mathcal{E}|$ | $S(|\mathcal{E}|)$ | Model Type | Parameters | Model Size |
|---------|-------------------|---------|-------|---------|------------|------------|------------|
| ER | - | - | 2000 | 5.85MB | Resnet32 | 0.46M | 1.76MB |
| REMIND | Feature replay | 5.85MB | - | - | Resnet32 | 0.46M | 1.76MB |
| DER | Logits | 0.08MB | 1974 | 5.77MB | Resnet32 | 0.46M | 1.76MB |
| ER-MIR | - | - | 2000 | 5.85MB | Resnet32 | 0.46M | 1.76MB |
| EWC | FI & Previous Model | 3.52MB | 794 | 2.33MB | Resnet32 | 0.46M | 1.76MB |
| BiC | Previous Model | 1.76MB | 1397 | 4.09MB | Resnet32 | 0.46M | 1.76MB |
| MEMO | Expanded Network | 1.33MB | 1542 | 4.51MB | Resnet32 | 0.46M | 1.76MB |
| L-SAR | Class-wise similarity & frequency of each sample | 8.52KB | 1997 | 5.84MB | Resnet32 | 0.46M | 1.76MB |

Table 4: Implementation details of total memory budget=7.6MB in CIFAR-10

| Methods | $\mathcal{B}$ Type | $S(|\mathcal{B}|)$ | $|\mathcal{E}|$ | $S(|\mathcal{E}|)$ | Model Type | Parameters | Model Size |
|---------|-------------------|---------|-------|---------|------------|------------|------------|
| ER | - | - | 2000 | 5.85MB | Resnet32 | 0.46M | 1.76MB |
| REMIND | Feature replay | 5.85MB | - | - | Resnet32 | 0.46M | 1.76MB |
| DER | Logits | 0.71MB | 1770 | 5.14MB | Resnet32 | 0.46M | 1.76MB |
| ER-MIR | - | - | 2000 | 5.85MB | Resnet32 | 0.46M | 1.76MB |
| EWC | FI & Previous Model | 3.52MB | 794 | 2.33MB | Resnet32 | 0.46M | 1.76MB |
| BiC | Previous Model | 1.76MB | 1397 | 4.09MB | Resnet32 | 0.46M | 1.76MB |
| MEMO | Expanded Network | 1.33MB | 1542 | 4.51MB | Resnet32 | 0.46M | 1.76MB |
| L-SAR | Class-wise similarity & frequency of each sample | 0.05MB | 1987 | 5.83MB | Resnet32 | 0.46M | 1.76MB |

Table 5: Implementation details of total memory budget=7.6MB in CIFAR-100

the episodic memory size. More specifically, DER needs additional storage which size is $|\mathcal{E}| \times d_l \times$ 4 bytes/float, where $d_l$ denotes logit dimension, which is 100 in CIFAR-100 and 10 in CIFAR-10.

L-SAR stores similarities between classes, the training frequency of each sample, and the trace of FIM for each layer. For instance, in CIFAR-10, L-SAR needs 400 bytes = 4 bytes/float $\times 10^2$ for saving class-wise similarities, 4 bytes/int $\times |\mathcal{E}|$ for saving frequency of each sample, and 4 bytes/float $\times n_l$ for saving trace of FIM for each layer, where $n_l$ is total number of layers. However, such additional memory cost is negligible compared to episodic memory or model parameters (only 0.1% of memory budget). We summarize implementation details of the total memory budget for each dataset in Tab.4, Tab.5, Tab.6, and Tab.7.

## A.4    IMPLEMENTATION DETAILS.

We use ResNet-32 (He et al., 2016) for CIFAR-10 and CIFAR-100, and use ResNet-18 as the network architecture for CLEAR-10 and ImageNet. We set the training hyperparameters as follows (Koh et al., 2022; Bang et al., 2021; Prabhu et al., 2020). For CIFAR-10, CIFAR-100, CLEAR-10, and ImageNet, we use batch size of 16, 16, 16, and 256, respectively, and Adam optimizer with LR of 0.0003 for all datasets and setup. For memory constraints, we used memory size of 7.6MB, 13.44MB, 25.12MB for CIFAR-10 and CIFAR-100, 617MB for CLEAR-10, 5.8GB for ImageNet.

For data augmentation, we apply RandAugment (Cubuk et al., 2020) to all CL methods. For hyperparameters, we set all the EMA ratios required for L-SAR to 0.01 for all datasets. For the values of $k$ and $T$ used in memory retrieval, we use $k = 4$ and $T = 0.125$ for all experiments. It is found by a hyperparameter search on CIFAR-10, and applied to other datasets without additional tuning since dataset-specific hyperparameter search is inappropriate for CL where the whole dataset is not given at once.

For L-SAR, we use memory-only training, where the training batch is retrieved from the episodic memory at every iteration. And we use the Greedy Balanced Sampling strategy (Prabhu et al., 2020) for memory sampling. We use $m = 4$ for all datasets and setups, where $m$ refers to the batch cycles where layer freezing is not applied..

## A.5    APPLYING ADAPTIVE LAYER FREEZING TO ATTENTION BASED MODEL

We investigate the effect of the proposed adaptive layer freezing not only in ResNet but also in attention-based models such as the Vision Transformer(ViT) (Dosovitskiy et al., 2020). We compare the freezing effects on models pretrained on ImageNet-1K and trained from scratch in Tab. 8

| Methods | $\mathcal{B}$ Type | S($|\mathcal{B}|$) | $|\mathcal{E}|$ | S($|\mathcal{E}|$) | Model Type | Parameters | Model Size |
|---------|--------------------|--------------------|-----------------|--------------------|------------|------------|------------|
| ER | - | - | 4,000 | 574MB | Resnet18 | 11.17M | 42.6MB |
| REMIND | Feature replay | 574MB | - | - | Resnet18 | 11.17M | 42.6MB |
| DER | Logits | 0.08MB | 3,999 | 573.92MB | Resnet18 | 11.17M | 42.6MB |
| ER-MIR | - | - | 4,000 | 574MB | Resnet18 | 11.17M | 42.6MB |
| EWC | FI & Previous Model | 85.2MB | 3,406 | 488.8MB | Resnet18 | 11.17M | 42.6MB |
| BiC | Previous Model | 42.6MB | 3,703 | 531.4MB | Resnet18 | 11.17M | 42.6MB |
| MEMO | Expanded Network | 31.95MB | 3,777 | 542.1MB | Resnet18 | 11.17M | 42.6MB |
| L-SAR | Class-wise similarity & frequency of each sample | 8.52KB | 3,999 | 573.86MB | Resnet18 | 11.17M | 42.6MB |

Table 6: Implementation details of total memory budget=616.6MB in CLEAR-10

| Methods | $\mathcal{B}$ Type | S($|\mathcal{B}|$) | $|\mathcal{E}|$ | S($|\mathcal{E}|$) | Model Type | Parameters | Model Size |
|---------|--------------------|--------------------|-----------------|--------------------|------------|------------|------------|
| ER | - | - | 20,000 | 2,870MB | Resnet18 | 11.17M | 42.6MB |
| REMIND | Feature replay | 2,870MB | - | - | Resnet18 | 11.17M | 42.6MB |
| DER | Logits | 74.3MB | 19,482 | 2,795.7MB | Resnet18 | 11.17M | 42.6MB |
| ER | - | - | 20,000 | 2,870MB | Resnet18 | 11.17M | 42.6MB |
| EWC | FI & Previous Model | 85.2MB | 19,406 | 2,784.8MB | Resnet18 | 11.17M | 42.6MB |
| BiC | Previous Model | 42.6MB | 19,703 | 2,827.4MB | Resnet18 | 11.17M | 42.6MB |
| MEMO | Expanded Network | 31.95MB | 19,787 | 2,838MB | Resnet18 | 11.17M | 42.6MB |
| L-SAR | Class-wise similarity & frequency of each sample | 4MB | 19,973 | 2,866MB | Resnet18 | 11.17M | 42.6MB |

Table 7: Implementation details of total memory budget=2,912.6MB in ImageNet

| Methods | Pretrained | | | From Scratch | | |
|---------|-------------------|-------------------|-----------|-------------------|-------------------|-----------|
| | $A_{\text{AUC}}\uparrow$ | $A_{last}\uparrow$ | TFLOPs $\downarrow$ | $A_{\text{AUC}}\uparrow$ | $A_{last}\uparrow$ | TFLOPs $\downarrow$ |
| Vanilla | 57.85±1.16 | 61.43±0.68 | 4,044.25 | 33.13±3.15 | 28.58±4.64 | 4,044.25 |
| + Adaptive Freezing | 58.70±1.58 | 60.73±1.52 | **3,466.42** | 33.04±3.44 | 30.44±6.22 | **3,926.65** |

Table 8: Effect of layer freezing in ViT. We used CIFAR-10 as the dataset, and the memory budget is 7.6MB

When using a pretrained model, the adaptive layer freezing reduces the computational cost by nearly 15% with minimal impact on $A_{AUC}$ and $A_{last}$, compared to the Vanilla training without freezing. Since pretrained models have already been sufficiently trained on a large dataset, the amount of information that the model will learn from the training data may be relatively small compared to training from scratch. Thus, it leads to the freezing of many layers by adaptive layer freezing. This not only reduces computational costs but also ensures high performance, since the model is updated only in truly informative batches, thus preserving the advantages of pretrained initialization.

In the case of training from scratch, the decrease in TFLOPs is significantly small compared to using a pretrained model, which implies that the layers did not freeze much. This is due to the large model capacity of ViT and the small number of training iterations in online CL, which leads to a severe underfitting of the model when training from scratch. Tab. 8 shows that when training from scratch, the accuracy only reaches around 30% even at the end of the training ($A_last$). Thus, since the model is not sufficiently trained yet, the adaptive layer freezing scheme tends to freeze fewer layers so that the model can learn more information. This shows that the proposed adaptive freezing method can indeed provide a reasonable freezing strategy.

### A.6 COMPARISON BETWEEN ADAPTIVE LAYER FREEZING AND NAIVE LAYER FREEZING

We compare the proposed adaptive layer freezing method with various naive freezing methods, in both Gaussian and disjoint setup in CIFAR-10. The results are summarized in Tab.9. Each freezing strategy chooses the number of frozen layers $n \in [0, L]$ where $L$ is the total number of layers, so that when $n \geq 1$, layer 1 to layer $n$ are frozen. The compared freezing strategies are: random freezing ($n$ is randomly selected from $[0, n_{\max}]$ every iteration for a fixed $n_max \in [0, L]$), constant freezing ($n$ is fixed initially) and linear freezing ($n$ increases linearly from 0 to $n_{\max}$ for a fixed $n_max \in [0, L]$).

All layer freezing strategies contribute to reducing computational costs. However, adaptive layer freezing has the least performance decrease. Note that the goal of layer freezing is not to freeze as much as possible but rather to save computational costs while preserving performance.

| Methods | Gaussian | | | Disjoint | | |
|---|---|---|---|---|---|---|
| | $A_{\text{AUC}} \uparrow$ | $A_{last} \uparrow$ | TFLOPs $\downarrow$ | $A_{\text{AUC}} \uparrow$ | $A_{last} \uparrow$ | TFLOPs $\downarrow$ |
| No Freezing | 64.60±0.83 | 72.43±0.38 | 171.94 | 79.10±0.44 | 71.77±0.57 | 171.94 |
| Random Freezing ($n_{\max} = 16$) | 63.14±0.51 | 70.47±1.15 | 150.56 | 77.69±0.51 | 69.30±1.77 | 150.62 |
| Random Freezing ($n_{\max} = 32$) | 61.79±0.54 | 69.84±0.54 | 122.99 | 77.31±0.17 | 68.89±0.64 | 120.92 |
| Constant Freezing ($n = 8$) | 60.91±0.80 | 67.70±0.83 | 147.12 | 74.99±0.24 | 65.89±0.50 | 147.12 |
| Constant Freezing ($n = 16$) | 53.59±0.60 | 57.31±0.90 | **109.48** | 67.64±0.61 | 55.59±0.29 | **109.48** |
| Linear Freezing ($n_{\max} = 16$) | 63.11±0.90 | 70.00±1.04 | 150.64 | 77.53±0.47 | 68.30±1.52 | 150.64 |
| Linear Freezing ($n_{\max} = 32$) | 62.06±0.90 | 66.95±2.31 | 120.83 | 75.69±0.77 | 64.49±1.22 | 120.83 |
| Adaptive Freezing (Ours) | 64.38±0.32 | 72.57±0.79 | 146.80 | 79.75±0.38 | 70.70±0.88 | 143.51 |

Table 9: Comparison between adaptive layer freezing and naive freezing in CIFAR-10. The memory budget is 7.6MB.

| Methods | CIFAR-10 | | | CIFAR-100 | | |
|---|---|---|---|---|---|---|
| | $A_{\text{AUC}} \uparrow$ | $A_{last} \uparrow$ | TFLOPs $\downarrow$ | $A_{\text{AUC}} \uparrow$ | $A_{last} \uparrow$ | TFLOPs $\downarrow$ |
| Vanilla | 77.10±0.58 | 70.26± 0.91 | 163.73 | 43.59±0.86 | 38.64±0.35 | 245.85 |
| + Freezing | 76.98±0.15 | 70.58±0.63 | **141.50** | 43.20±0.95 | 38.44±0.27 | **225.92** |
| + Similarity Aware Retrieval | 79.10±0.44 | 71.77±0.57 | 171.94 | 45.61±1.06 | 39.68±0.66 | 257.91 |
| + Similarity Aware Retrieval & Freezing (L-SAR) | 79.75±0.38 | 70.70±0.88 | 143.51 | 45.00±1.28 | 39.39±0.62 | 228.14 |

Table 10: Benefits of the proposed components of our method in CIFAR-10 and CIFAR-100 for disjoint task setup. The memory budget is 7.6MB for CIFAR-10 and 13.44MB for CIFAR-100. CIFAR-10 We train for 1 iter per sample for CIFAR-10 and 1.5 iter per sample for CIFAR-100.

## A.7 ADDITIONAL ABLATION STUDY

In addition to Gaussian task setup, we ablate the model to investigate the benefit of each of the proposed components by using CIFAR-10 and CIFAR-100 in the disjoint task setup, and summarize the results in Table 10. Summing up the effect of each component, L-SAR outperforms the baseline while using fewer FLOPs than the baseline, each by a noticeable margin.

## A.8 RESULTS USING VARIOUS MEMORY BUDGETS IN CIFAR-10

| Methods | Memory Size | | | | | |
|---|---|---|---|---|---|---|
| | 7.6 MB | | 13.44 MB | | 25.12 MB | |
| | $A_{\text{AUC}} \uparrow$ | $A_{last} \uparrow$ | $A_{\text{AUC}} \uparrow$ | $A_{last} \uparrow$ | $A_{\text{AUC}} \uparrow$ | $A_{last} \uparrow$ |
| ER (Rolnick et al., 2019) | 57.69±0.33 | 69.53±0.10 | 58.10±0.97 | 72.42±0.26 | 57.95±0.37 | 73.11±0.50 |
| REMIND (Hayes et al., 2020) | 56.06±0.45 | 58.91±0.99 | 55.74±0.24 | 60.97±1.14 | 56.14±0.68 | 61.61±0.93 |
| DER++ (Buzzega et al., 2020) | 53.01±0.15 | 68.35±0.40 | 55.37±0.44 | 68.57±2.33 | 55.72±0.51 | 68.84±1.90 |
| ER-MIR (Aljundi et al., 2019a) | 47.83±0.15 | 58.41±1.18 | 47.35±0.76 | 58.03±2.51 | 47.88±1.09 | 58.41±3.84 |
| EWC (Kirkpatrick et al., 2017) | 54.71±0.38 | 61.45±3.18 | 56.26±0.34 | 68.62±1.63 | 56.60±0.29 | 69.59±1.21 |
| BiC (Wu et al., 2019) | 48.59±1.25 | 60.67±0.99 | 47.80±0.75 | 62.00±1.90 | 45.30±1.35 | 58.21±1.86 |
| MEMO (Zhou et al., 2023) | 49.56±0.66 | 45.64±3.80 | 49.50±1.06 | 56.52±1.71 | 48.88±1.35 | 57.70±3.55 |
| L-SAR (Ours) | **63.46±0.29** | **71.64±0.20** | **63.56±0.40** | **74.66±0.41** | **63.07±0.23** | **77.36±0.40** |

Table 11: Comparison of accuracy for different memory sizes for Gaussian data Stream in CIFAR-10. The computational budget is fixed as 128.95 FLOPs

The results obtained using various memory budgets in CIFAR-10 are shown in Tab. 11. As in CIFAR-100, our method outperforms other methods in all tested memory budgets, further showing that our method is robust across various memory constraints.

## A.9 COMPATIBILITY OF L-SAR ACROSS DIVERSE NETWORK ARCHITECTURES

L-SAR can be applied to any feedforward neural network, as long as layers can be defined, including CNNs and Vision Transformers (Dosovitskiy et al., 2020). Since our layer freezing methods require evaluating the information gained and FLOPs used by individual layers, a network should be dissected into layers. However, our proposed adaptive layer freezing cannot apply to recurrent neural networks (Sherstinsky, 2020) since the gradient of a layer affects not only the preceding layers but also the subsequent layers (Rotman & Wolf, 2021), while in the feedforward network, the gradient of a layer influences only the preceding layers.

## A.10 Details about $A_{\text{AUC}}$

Recent studies (Pellegrini et al., 2020; Caccia et al., 2022; Banerjee et al., 2023; Ghunaim et al., 2023) suggest that having good inference performance at any intermediate time points during training is important for CL. To evaluate intermediate performance during training, Koh et al. (2022) proposed $A_{\text{AUC}}$, which measures the area under the curve of average accuracy. In contrast to $A_{last}$ or $A_{avg}$ which measures performance only at the end of the task (*i.e.*, after sufficient training), $A_{AUC}$ consistently measures the performance over the course of training. If two methods reach the same accuracy at the end of a task but one method converges faster than the other, their $A_{last}$ and $A_{avg}$ would be equal, but the faster model would show higher $A_{AUC}$. Thus, how fast the model adapts to the new task is reflected in $A_{AUC}$.

## A.11 Details about estimation of Fisher Information trace

To check how accurate our Fisher Information trace estimate is, we ran an experiment comparing the Fisher Information trace estimated with i) a batch size of 16 once every four steps and ii) a batch size of 64 for every step (*i.e.*, 16 times bigger sample size) on CIFAR-10 Gaussian task setup. We use ResNet-32 as the backbone and show the trace of the Fisher Information of the last layers for each block, *i.e.* layers 8, 16, 24, and 32. From the result in Fig. 5, we observe that the estimation with i) a batch size of 16 once every four steps does not deviate much from the estimation with ii) a batch size of 64 for every step, showing that our estimation is reasonably accurate.

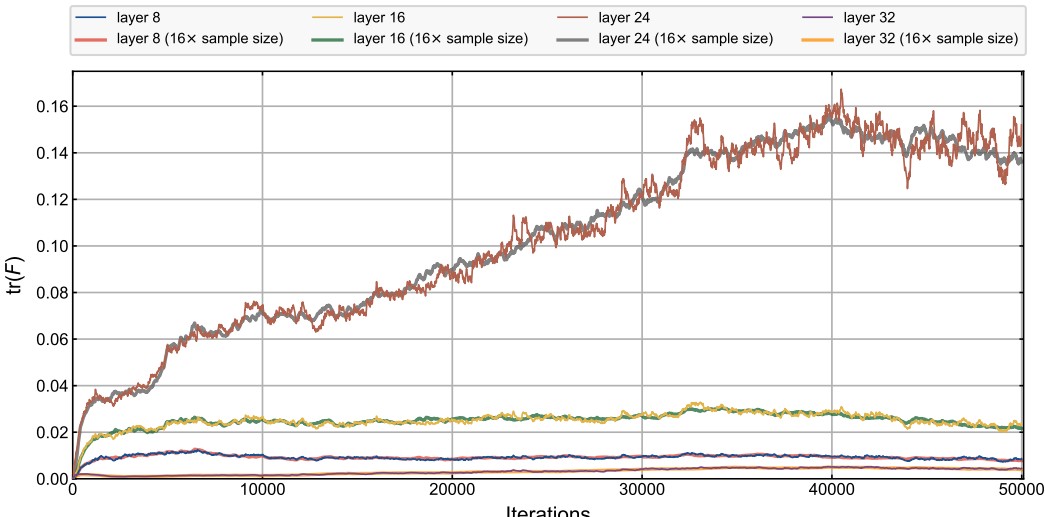

Figure 5: The estimated trace of Fisher Information for layers 8, 16, 24, and 32 of ResNet-32 on CIFAR-10 Gaussian Task setup, comparing the estimation used in L-SAR and the estimation with a 16 times bigger sample size.

