# OpenReview forum: "Budgeted Online Continual Learning by Adaptive Layer Freezing and Frequency-based Sampling"
_ICLR.cc/2024/Conference — Submitted to ICLR 2024_

### Official Review · Reviewer_7DQF · 2023-10-30

**Soundness:** 2 fair
**Presentation:** 3 good
**Contribution:** 2 fair
**Rating:** 5
**Confidence:** 2

**Summary:**

This work considers the online continual learning setting with computational restrictions. The authors propose an adaptive layer freezing approach based on the information gain per iteration measured by the Fisher information and other heuristic design. Additionally, to guarantee adequate information acquisition by the network, they suggest a frequency-based, similarity-aware retrieval schedule.

This work aims at improving the computational efficiency without an extensive focus on adapting specific neural architectures. Extensive experiments are provided to substantiate the proposed algorithmic approach.

**Strengths:**

The writing and presentation are clear, with detailed comparisons to prior research. This study benchmarks computational resources in terms of FLOPS, offering a relatable metric to appreciate its contributions. The paper carries out extensive experiments to support the results.

**Weaknesses:**

See the Questions.

**Questions:**

I haven't observed an "unfreezing" step within the proposed algorithm. Given that the focus is on online continual learning, how does the algorithm account for potential shifts in data distributions?

The constraints on computational resources could potentially lead to suboptimal performance. How are the hyperparameters adjusted to ensure a dynamic balance between efficiency and accuracy?

The application of these methods across diverse network architectures remains unclear to me. Would there be a requirement to rescale the information gain, especially considering that wider layers might inherently convey more information? Additionally, I remain uncertain about how this method can be applied to specialized structures like attention mechanisms. Could you provide some clarity on this?

---

> ### Author Response · Authors · 2023-11-15
> **Answers to the questions of Reviewer 7DQF**
>
> We thank reviewer 7DQF for the encouraging remarks on clear presentation and supporting extensive experiments. We address your concerns as follows.
>
> > I haven't observed an "unfreezing" step within the proposed algorithm.
>
> $\to$ There is an unfreezing step at every iteration in our algorithm since the layer freezing method adaptively freezes/unfreezes the layers at each iteration.
>
> > Given that the focus is on online continual learning, how does the algorithm account for potential shifts in data distributions?
>
> $\to$ Great question! When the data distribution shifts, only a few layers are frozen due to the high Fisher information for batches containing the data from the new data distribution. As the training progresses and the Fisher information computed from batches decreases, *i.e.*, the distribution remains constant, the number of freezing layers increases. In sum, our proposed freezing method adaptively freezes by the amount of information of the batch, and it adapts to the distribution shifts.
>
> > The constraints on computational resources could potentially lead to suboptimal performance. How are the hyperparameters adjusted to ensure a dynamic balance between efficiency and accuracy?
>
> $\to$ We did not adjust the hyperparameters such as m, which is introduced in Eq.4 and means the period where layer freezing is not applied, for varying computational constraints. In CL, it is unrealistic to perform separate hyperparameter searches for each experimental setup since the model would not know which data it will encounter in the future. Thus, we conducted a hyperparameter search on CIFAR-10, Gaussian task setup, with the computational budget of 163.73 TFLOPs (1 iteration/sample for baseline), and used the same hyperparameter for all datasets, setup, and constraints. Rather than adjusting hyperparameters, our approach seeks to dynamically balance efficiency and accuracy by the adaptive freezing method, which automatically adjusts the degree of freezing to maximize information gain per computation. It would be an interesting future work to dynamically adjust hyperparameters depending on the amount of budgets without performing a budget-specific hyperparameter search.
>
> > The application of these methods across diverse network architectures remains unclear to me.
> >> Would there be a requirement to rescale the information gain, especially considering that wider layers might inherently convey more information?
>
> $\to$ No, rescaling is not required even in the wider layers. It is because the proposed approach determines freezing by information gained per FLOPs cost rather than information gain alone. While it is true that wider layers convey more information, they also cost more FLOPs for forward and backward passes. In our freezing criterion, the two factors balance out, thus rescaling is not required.
>
> >> Additionally, I remain uncertain about how this method can be applied to specialized structures like attention mechanisms. Could you provide some clarity on this?
>
> $\to$ Thank you for the very interesting question. We believe that our proposed method can be applied to various feedforward networks with layer structures including CNNs and Transformers, since the only architecture-specific part of our method is dissecting the Fisher Information and FLOPs into layers for freezing criterion. To empirically verify the effectiveness of our method on different network structures, we applied our proposed adaptive layer freezing method to ViT, an attention-based model, and reported the results in appendix Sec.A.5 of the revision. When adaptive layer freezing was applied to the pre-trained ViT, it reduced the computational costs by 15%, with negligible change in accuracy (+0.85% in $A_{AUC}$, -0.67% in $A_{last}$). In from-scratch training, layer freezing resulted in accuracy (-0.09% in $A_{AUC}$, +1.86% in $A_{last}$), with a 3% reduction in computational cost.
>  While L-SAR can be applied to any feedforward network with a layer structure, it can not be applied to recurrent neural networks. In a recurrent neural network, the backward pass goes through multiple loops over the whole network, so we cannot stop gradient calculation for specific layers without affecting the whole network.

---

> > ### Comment · Reviewer_7DQF · 2023-11-22
> > **Response to Rebuttal**
> >
> > I want to thank the authors for their comprehensive response. I decide to maintain the score.

---

> > > ### Author Response · Authors · 2023-11-22
> > > **Reply to the official comment by reviewer 7DQF**
> > >
> > > Thank you very much for your response. Would you be able to provide specific reasons for your decision so that we have a chance to clarify the arguments and revise the manuscript?

---

### Official Review · Reviewer_htG8 · 2023-10-31

**Soundness:** 2 fair
**Presentation:** 2 fair
**Contribution:** 2 fair
**Rating:** 5
**Confidence:** 2

**Summary:**

This work is an improvement of the method to solve the online continual learning problem with budgeted computational power. The proposed method, based on measuring the computational cost in FLOPs and storage cost in bytes, consists of two designs: adaptive layer freezing and frequency-based sampling. Adaptive layer freezing uses Fisher Information to determine the layers to skip updating for less informative batches. Frequency-based sampling estimates the similarity between categories and maintains a relatively constant use frequency across samples. This work compares the “last accuracy” (A[last]) and the “area-under-the-curve of accuracy” (A[AUC]) of both Gaussian task and disjoint task of CIFAR-10, CIFAR-100, ImageNet under certain computational budget and claims an outperforming improvement against seven other existing algorithms.

**Strengths:**

+ This paper targets the emerging continual learning.
+ The proposed method is supported by strong statistics theories.
+ This work carries out extensive evaluation and comparison among different algorithms.

**Weaknesses:**

- It would be helpful to further improve the presentation of this paper.
- There are occasional grammatical, phrasing, and spelling issues.
- It would be helpful to further highlight the difference between this work and existing efforts.
- Some claims in this paper require more supporting evidence.

**Questions:**

1. Section 4.1: Regarding the claim “A[AUC] is a suitable metric …”, is there any evidence or findings to support this claim?
2. Figure 3: L-SAR chooses slightly lower GFLOPs per sample than other methods. It would be helpful to explain the reason leading to this decision.
3. Considering the contribution claim of measuring computational budget in FLOPs and storage budget in bytes, how does this work compare against prior work?
4. The term “logit” seems ambiguous and can mean multiple things. Does it refer to “the unnormalized output vector of a classification model before the softmax operator” (a definition specific to TensorFlow)?
5. With the claim in the abstract, “the new and advanced algorithms often perform worse than simple baselines under the same budget”, it would be better to explain the abnormality in detail. Are these algorithms running on a budget on par with what they are designed for? What are the situations in which those algorithms excel?

---

> ### Author Response · Authors · 2023-11-15
> **Answers to the questions of Reviewer htG8**
>
> We thank reviewer htG8 for the encouraging remarks on strong statistical theories, supporting extensive experiments. We address your concerns as follows.
>
> > Section 4.1: Regarding the claim “A[AUC] is a suitable metric …”, is there any evidence or findings to support this claim?
>
> $\to$ In contrast to $A_{last}$ or $A_{avg}$ which measures the performance only at the end of the task (*i.e.*, after sufficient training), $A_{AUC}$ measures the performance over the course of training. Thus, when the model quickly adapts to the new task,  $A_{AUC}$ increases. For example, if two methods reach the same accuracy at the end of a task but one method converges faster than the other, their $A_{last}$ and $A_{avg}$ would be equal but the faster model would show higher $A_{AUC}$. This effect is also observed in Koh et al., 2023, as their proposed distillation method shows a 5.23% gain in $A_{last}$ over the baseline, but the gain in $A_{AUC}$ is only 1.67% for CIFAR-10 Gaussian scheduled task setup. While distillation prevents forgetting and allows the model to converge to higher accuracy than baseline, leading to high $A_{last}$, distillation hinders models to learn new tasks since it regularizes the update of previously learned representation. Thus, the gain in $A_{AUC}$ is smaller than $A_{last}$ since the drawback of slow learning decreases $A_{AUC}$.
>
> > Figure 3: L-SAR chooses slightly lower GFLOPs per sample than other methods. It would be helpful to explain the reason leading to this decision.
>
> $\to$ The reason that L-SAR’s GFLOPs/sample is lower than other baseline methods is that it adaptively freezes layers at every training batch. As GFLOPs/sample of other baselines can be measured accurately, we set the iteration/sample for each method to ensure having the same GFLOPs/sample. However, since L-SAR adaptively freezes layers at every training batch, it is hard to accurately set iteration/sample of L-SAR to have the same GFLOPs/sample with other baselines. Even though we set the iteration/sample of L-SAR as ER, which is a simple baseline, L-SAR has lower GFLOPs/sample than other methods due to adaptive layer freezing.
>
> > Considering the contribution claim of measuring computational budget in FLOPs and storage budget in bytes, how does this work compare against prior work?
>
> $\to$ (Prabhu et al., 2023, Ghunaim et al., 2023) consider the relative FLOPs, which is the ratio of the FLOPs used by each method to the FLOPs of ER,  for the computational budget. Zhou et al., 2023 use byte for the storage budget. We are the first to apply both for the resource budget. The relative FLOPs is not a proper metric when the required FLOPs for training are dynamically changing.
>
> > The term “logit” seems ambiguous and can mean multiple things. Does it refer to “the unnormalized output vector of a classification model before the softmax operator” (a definition specific to TensorFlow)?
>
> $\to$ Yes, you are correct. We follow the definition of logit by Buzzega et al, 2020. We add the definition of logit in Sec.1 in the revision. Thank you!
>
>
> > With the claim in the abstract, “the new and advanced algorithms often perform worse than simple baselines under the same budget”, it would be better to explain the abnormality in detail.
> > > Are these algorithms running on a budget on par with what they are designed for?
>
> $\to$ Yes for the online CL methods (ER, DER, MIR and REMIND), but not for the offline CL methods (EWC, BiC, MEMO). Each online CL method uses different amounts of computation and memory budget, so we have evaluated them on a wide range of computation and memory budgets in Sec.4.2 so that our experiments cover the budgets each algorithm is designed for. Meanwhile, ER, a commonly used baseline in online CL, outperforms other existing CL methods under the computation and memory budget same as the budgets the other CL methods use. For example, MIR is an upgrade of ER by using a better retrieval strategy and outperforms ER, compared under the same number of iterations. However, MIR uses approximately 3.5 times more computations than ER. When ER is also given 3.5 times more computations, ER outperforms MIR.
>
> Offline CL methods, however, are typically designed to use a much larger computational budget than online CL methods. Although we have implemented them based on ER to fit the online scenarios, the budget is not comparable with what they were designed for.
>
>
> > > What are the situations in which those algorithms excel?
>
> $\to$ The computation-heavy methods such as MIR and BiC may outperform ER when compared under the same number of iterations. However, for a fair comparison with the computational budget, we compare the methods under the same FLOPs instead of iterations. Also, the offline CL methods, namely EWC, BiC, and MEMO, may excel in an offline scenario. We focus on a more realistic and challenging online CL scenario, so offline performance was out of the scope.

---

> > ### Comment · Reviewer_htG8 · 2023-11-22
> >
> > I sincerely appreciate the authors' careful response to my questions. Most of my questions are addressed. However, after reading others' comments, I want to maintain my score.

---

> > > ### Author Response · Authors · 2023-11-22
> > > **Reply to the official comment by reviewer htG8**
> > >
> > > Thank you very much for your response. Would you be able to provide specific reasons for your decision so that we have a chance to clarify the arguments and revise the manuscript?

---

### Official Review · Reviewer_sCXL · 2023-11-02

**Soundness:** 2 fair
**Presentation:** 2 fair
**Contribution:** 2 fair
**Rating:** 5
**Confidence:** 4

**Summary:**

This paper claims two contributions. First, it makes the case that in order to fairly compare different continual learning algorithms, we need to compare them under the same computational and total-memory constraints. Second, it proposes a novel approach, which combines adaptive layer freezing (to reduce computation) and targeted memory retrieval for more efficient learning.

**Strengths:**

The main strength of this paper is that it makes the case that we need to start taking into account the computational demands of continual learning algorithms. I fully agree with the authors here.

**Weaknesses:**

- In my opinion, Section 3 needs a lot of work before the paper is ready for publication, because it fails to adequately and clearly explain the two components of the proposed approach. I have personally read this section four times, inlcuding a lot of back-and-forth to the appendix, in order to fully understand the proposed approach. I also think that the pseudocode is too important, and has to be included in the main paper. The same holds for the information discussed in Sections A.2 and A.3.
- I am not sure I agree with the way the authors measure computational cost. According to my understanding, neural network training can be split in three stages: i) the forward pass, during which we calculate layer activations; ii) the backward pass, during which we calculate layer gradients; iii) the update of the weights. I think that the paper might conflate the backward pass with the weight updates. The proposed approach computes all the layer gradients regardless of the adaptive layer freezing. Hence, the backward pass is always completed in full, and the computational savings might be much less than what they appear at first glance (please correct me if I'm wrong).
- I think you should include another section in the appendix, where you discuss in detail the memory sizes (number of instances) you use in all your experiments. I would also appreciate a more detailed description of the calculations behind the memory sizes (with examples).
- I found the ablation to be incomplete, as it only includes the Gaussian setup (but not the disjoint), and it does not examine the Vanilla+Freezing approach (i.e., with random retrieval).
- The paper does not include any wall-clock time comparison to further reinforce their comparison with respect to FLOPs.
- There are multiple issues with the references (i.e., duplicates, references with incorrect publication status).

**Questions:**

- Could you specify exactly which computations are counted in the L-SAR computational budget?
- Could you explain exactly how you calculate $A_{\text{AUC}}$? In particular, are you using the entire test set at every evaluation point, and how often do you evaluate?
- I could not find any mention of the number of runs for each experiment.
- Did you use a test to decide which results are a statistically significant improvement? In all tables, you always bold the highest mean accuracy of each column, but I think that you should only bold when the difference is statistically significant.
- Do you train the model only with data sampled from memory or do you also use stream observations (e.g., like in ER or MIR)?
- Did you compare your freezing strategy with a naive freezing strategy (e.g., one where you determine the freezing cutoff layer at random)?
- Why do you use only the gradient of the last feature layer in Eqs. (4) and (6)?
- You write "We evaluate the methods in [...] a newly proposed Gaussian task setup for boundary-free continuous data stream (Wang et al., 2022d; Koh et al., 2022)." However, Wang et al. adopt the setting of [1], while Koh et al. propose a periodic setup that is different from [1]. Could you clarify which setup you are using?
- In Algorithm 1, line 22, there is an acronym "B-FIUC" which I did not encounter anywhere else in the paper. Could you explain what it is?

[1] Murray Shanahan, Christos Kaplanis, and Jovana Mitrovic. Encoders and ensembles for task-free continual learning. arXiv preprint arXiv:2105.13327, 2021.

---

> ### Author Response · Authors · 2023-11-15
> **Answers to the questions of Reviewer sCXL (1/3)**
>
> We thank reviewer sCXL for the encouraging remarks on motivation. We address your concerns as follows.
>
> > In my opinion, Section 3 needs a lot of work before the paper is ready for publication because it fails to adequately and clearly explain the two components of the proposed approach. I have personally read this section four times, including a lot of back-and-forth to the appendix, in order to fully understand the proposed approach. I also think that the pseudocode is too important, and has to be included in the main paper. The same holds for the information discussed in Sections A.2 and A.3.
>
> $\to$ Sincere apology for the clarity of Sec. 3. We revise and rearrange Sec.3 and Appendix in the revision to improve the clarity. First, to contain important details of the algorithm as suggested, we have relocated the contents in Sec. A.2 and Sec. A.3 from the appendix to Sec. 3. Second, we made the motivational arguments more concise, and moved the derivation of the equation for computing batch-wise $(I/C)$ to Sec. A.1. Last, we rephrase many sentences for better readability. However, we could not move the full pseudocode to the main paper due to space. Instead, we added some details that were previously only given in the pseudocode, such as estimating the trace of Fisher Information instead of the full matrix and using only the unfrozen layers for calculating cosine similarity when estimating class similarity.
>
> > I am not sure I agree with the way the authors measure computational cost. According to my understanding, neural network training can be split in three stages: i) the forward pass, during which we calculate layer activations; ii) the backward pass, during which we calculate layer gradients; iii) the update of the weights. I think that the paper might conflate the backward pass with the weight updates. The proposed approach computes all the layer gradients regardless of the adaptive layer freezing. Hence, the backward pass is always completed in full, and the computational savings might be much less than what they appear at first glance (please correct me if I'm wrong).
>
>
>
> $\to$ As you mentioned, we considered the neural network training by the three stages. But we did not conflate the backward pass and the weight updates. Specifically, we do not compute the gradient for the frozen layers, so it reduces the cost of both the backward pass (*i.e.*, gradient calculation) and weight update. For $n^*$ that maximizes (Eq. 4), we freeze layer 1 to layer $n^*$, then we stop the backward pass at layer $n^*+1$. Thus, we do not calculate the gradient for layers 1 to $n^*$.
>
> Next, we will go through the other uses of gradients in our algorithm, to clarify that we do not always use all the layer gradients. In our method, the gradient is used to estimate (i) Fisher Information and (ii) Class Similarities. For (i), even though Fisher Information requires gradients for all layers, we only update the Fisher Information once every 4 iterations, in which we do not freeze the model. For the other 3 iterations where we use adaptive freezing, we do not update the Fisher Information, *i.e.*, backward pass only occurs in unfrozen layers. For (ii), we calculate class similarities only using the gradients of the unfrozen layers. Therefore, gradients of the frozen layers are not used, so we may safely skip the gradient calculations for those layers. We apologize that some of these important details were only included in the appendix and pseudocode, and have revised Sec.3 to contain them.
>
> > I think you should include another section in the appendix, where you discuss in detail the memory sizes (number of instances) you use in all your experiments. I would also appreciate a more detailed description of the calculations behind the memory sizes (with examples).
>
> $\to$ Thank you for the valuable suggestion! We add details of memory constraints in Sec. A.5 of the revision with examples of calculating memory budget as suggested.
> Providing an example of the memory budget calculation process with a total memory budget of 7.6MB and a ResNet-32 model type in CIFAR100, ER can store up to 2000 instances in episodic memory. On the contrary, EWC is limited to storing only 794(=2000 - 2$\times$603) instances, since EWC requires storing previous model parameters and the parameter-wise Fisher Information for all parameters and model size of ResNet-32 is same as 603 CIFAR100 instances. Similarly, BiC requires storing previous learned models, thus it can store only 1,397(=2,000 - 603) instances in episodic memory. If it is still not clear, please let us know.

---

> ### Author Response · Authors · 2023-11-15
> **Answers to the questions of Reviewer sCXL (2/3)**
>
> > I found the ablation to be incomplete, as it only includes the Gaussian setup (but not the disjoint), and it does not examine the Vanilla+Freezing approach (i.e., with random retrieval).
>
> $\to$ As suggested, we add ablation results in the disjoint task setup in Tab.9 of the revision. Similar to the Gaussian task setup, we observe notable performance improvement by our Layer Freezing and Similarity-Aware Retrieval in disjoint task setup. This is because L-SAR  prioritizes the retrieval of less used data, possibly leading to rapid adaptation to novel tasks. Furthermore, it saves computational cost by decreasing the number of frozen layers when data of novel task encounters, while increasing the number of frozen layers when training with data that have already been sufficiently trained.
> For the second comment about Vanilla+Freezing, we compare the random retrieval + Adaptive Layer Freezing approach in the Tab.3 of the revision. We observed a significant decrease in computational cost (in FLOPs) through freezing not only in our proposed similarity-aware retrieval but also in random retrieval.
>
> > The paper does not include any wall-clock time comparison to further reinforce their comparison with respect to FLOPs.
>
> $\to$ We also considered the wall-time clock as one of the metrics for the computational costs. However, when running experiments, we encountered significant variations in wall-time clock results due to system factors such as interference with other tasks and scheduling, even for the same experiments. This high variability made it impractical to use the wall-time clock as a reliable computational metric as we mentioned in Sec.1.
> In addition, previous work (Prabhu et al., 2023) also mentioned the problem of the wall-time clock, that it can vary according to hardware (*e.g.*, CPU and GPU) or suboptimal implementations when comparing methods. Therefore, for a fair comparison of all the methods without any dependency on environments, we used FLOPs as a metric of computational cost.
>
> > There are multiple issues with the references (*i.e.*, duplicates, references with incorrect publication status).
>
> $\to$ We sincerely apologize for the mistakes. We thoroughly revise the bib file and fix the mistakes including duplicated references.
>
> > Could you specify exactly which computations are counted in the L-SAR computational budget?
>
> $\to$ We count all the computations required for training in each CL method, including forward pass, backward pass (gradient calculation), model weight update, and additional computations (*e.g.*, calculating the cosine similarity for retrieval in L-SAR).
>
>
>
> > Could you explain exactly how you calculate $A_\text{AUC}$? In particular, are you using the entire test set at every evaluation point, and how often do you evaluate?
>
> $\to$ To calculate $A_\text{AUC}$, we use the entire test set for the class seen so far as the evaluation set, and evaluate the model at every 100 samples for CIFAR10, CIFAR100 and CLEAR10, and at every 8,000 samples for ImageNet. We add it to Sec.4.1.
>
> > I could not find any mention of the number of runs for each experiment.
>
> $\to$ For each experiment, we ran 3 random seeds and calculated avg and standard deviation. we add this description in Sec.4.1. Thank you!

---

> ### Author Response · Authors · 2023-11-15
> **Answers to the questions of Reviewer sCXL (3/3)**
>
> > Did you use a test to decide which results are a statistically significant improvement? In all tables, you always bold the highest mean accuracy of each column, but I think that you should only bold when the difference is statistically significant.
>
> $\to$ Great suggestion! We conduct Welch's $t$-test with a significance level of 0.05. Results with the highest performance and statistically significant are highlighted in bold, while underlined when they are not statistically significant. We add a description in Sec.4.1 of the revision. Thank you.
>
> > Do you train the model only with data sampled from memory or do you also use stream observations (e.g., like in ER or MIR)?
>
> $\to$ We use memory-only training, where the training batch is retrieved from the episodic memory at every iteration. And we use the Greedy Balanced Sampling strategy (Prabhu et al., 2020) for memory sampling. We add this in Sec.A.4 of the revision.
>
> > Did you compare your freezing strategy with a naive freezing strategy (e.g., one where you determine the freezing cutoff layer at random)?
>
> $\to$ Thank you for the interesting suggestion! We added the result of comparing adaptive layer freezing and naive freezing strategies such as freezing randomly selected layers, consistently freezing until a specific layer, and linearly increasing the freezing layers as the training progresses in Tab.9 (Sec.A.6). Compared to the naive freezing strategy, our proposed adaptive layer freezing effectively freezes layers with minimal performance degradation from not freezing layers.
>
> > Why do you use only the gradient of the last feature layer in Eqs. (4) and (6)?
>
> $\to$ It is for reducing the cost of computing gradients of all layers, following (Koh et al., 2023) (Sec. A.1). Since the gradient across all layers is proportional to the last layer feature’s gradient by the chain rule (*i.e.*, if the last layer gradient is scaled by a factor of 2, then the full gradient is also scaled by the factor of 2), it can be used to predict the magnitude of the full gradient as in (Koh et al., 2023).
>
> > You write "We evaluate the methods in [...] a newly proposed Gaussian task setup for boundary-free continuous data stream (Wang et al., 2022d; Koh et al., 2022)." However, Wang et al. adopt the setting of Shanahan et al. while Koh et al. propose a periodic setup that is different from Shanahan et al.. Could you clarify which setup you are using?
>
> $\to$ We use a non-periodic Gaussian task setup, as in (Wang et al., 2022d). We add (Shanahan et al., 2021) as suggested. (Koh et al., 2022) was included in reference because in addition to the periodic setup, they defined and used non-periodic Gaussian task setup.
>
> > In Algorithm 1, line 22, there is an acronym "B-FIUC" which I did not encounter anywhere else in the paper. Could you explain what it is?
>
> $\to$ It is $(I_\text{batch}/C)_n(z_t)$ (revised Algorithm 1 in Sec.A.2 of the revision). Sorry for the mistake.

---

> ### Comment · Reviewer_sCXL · 2023-11-21
> **Response to Rebuttal**
>
> I want to thank the authors for their detailed response. Having read it, here are some things I am still wondering about:
> - For the calculation of the Fisher Information trace, do you use the same batch size as the one in training steps (i.e., 16, 16, 16, 256)? I am wondering how accurate the Fisher Information trace estimate is if you use, for instance, a batch size of 16 once every four steps (as you write in your response).
> - Regarding my question about which computations are included in the computational budget of L-SAR, do you also include the cost of calculating the Fisher Information trace once every four training steps?
> - It seems to me that when you use Similarity-Aware Retrieval to sample a batch from the memory, you cannot use the same batch to compute the Fisher Information trace, since that batch is not sampled uniformly at random from the memory and the trace would be calculated incorrectly. What do you think?
> - You write in the updated manuscript "We argue that it allows us to dynamically adjust the degree of freezing based on the learning capacity of the layers and the batch informativeness." How can you tell how informative each individual batch is, if you only update the Fisher Information trace only once every four training steps?
> - Unfortunately, I don't think the comparison to Random and Linearly-Increasing Freezing is fair since they use up less computation than Adaptive Freezing. For example, in the case of Random Freezing you could maybe select $n \in [n_\text{min}, L]$, where $n_\text{min}$ is set so that the total cost is equal to your Adaptive Freezing, and you could also do something similar in Linearly-Increasing Freezing.

---

> > ### Author Response · Authors · 2023-11-22
> > **Additional response by the authors to the reviewer sCXL's response (2/2)**
> >
> > > You write in the updated manuscript "We argue that it allows us to dynamically adjust the degree of freezing based on the learning capacity of the layers and the batch informativeness." How can you tell how informative each batch is, if you only update the Fisher Information trace only once every four training steps?
> >
> > $\to$ We approximate the information in each batch by multiplying the Fisher Information (FI) for the overall training distribution (updated every four training steps as you mentioned) with the squared gradient magnitude of the batch divided by the average of squared gradient magnitude over training distribution (*i.e.*, multiple batches). In equations, $F_\text{batch}(\theta, z_t) = \frac{|\nabla \ell(z_t)|^2}{\mathbb{E}_z\left[|\nabla \ell(z)|^2\right] }\cdot F(\theta)$,
> >
> > where $\theta$ is the model parameters, $F(\theta)$ is the FI for the overall training distribution, $F_\text{batch}(\theta, z_t)$ is the FI for the current training batch $z_t$, and $z$ represents possible training batches in the training distribution, and $\ell$ is the loss function. This approximation is based on the definition of FI (Eq.1), where its diagonal components are the expectation of squared gradients, which implies that the larger the squared gradient, the larger the FI in the batch.
> >
> > Additionally, note that to assess the gradient magnitude without full backward pass, we only use the last layer’s gradient magnitude for computational efficiency, since gradients of previous layers are proportional to the gradient of the last layer by the chain rule.
> >
> > > Unfortunately, I don't think the comparison to Random and Linearly-Increasing Freezing is fair since they use up less computation than Adaptive Freezing. For example, in the case of Random Freezing you could maybe select $n \in [n_\text{min}, L]$, where $n_\text{min}$ is set so that the total cost is equal to your Adaptive Freezing, and you could also do something similar in Linearly-Increasing Freezing.
> >
> > $\to$ Fair point. As you suggested, we modified $n_\text{min}$ to set the computational cost equal to our adaptive layer freezing, and our adaptive layer freezing outperforms other naive freezing strategies in the same computational cost. We updated the results in Tab.9 of the revision.
> >
> > **Table A Comparison between adaptive layer freezing and naive freezing in CIFAR-10**
> > | Freezing Method | $A_\text{AUC}~\uparrow$ | $A_{last}~\uparrow$ | TFLOPs | $A_\text{AUC}~\uparrow$ | $A_{last}~\uparrow$ | TFLOPs |
> > |----------------------|---------|---------|---------|---------|---------|---------|
> > | Random Freezing | 63.14$\pm$0.51 | 70.47$\pm$1.15 | 150.56  | 77.69$\pm$0.51 | 69.30$\pm$1.77 | 150.62 |
> > | Linearly Increasing Freezing | 63.11$\pm$0.90 | 70.00$\pm$1.04 | 150.64 | 77.53$\pm$0.47 | 68.30$\pm$1.52 | 150.64 |
> > | Adaptive Freezing (Ours) | 64.38$\pm$0.32 | 72.57$\pm$0.79 | 146.80 | 79.75$\pm$0.38 | 70.70$\pm$0.88 | 143.51 |

---

> > > ### Comment · Reviewer_sCXL · 2023-11-22
> > > **Response**
> > >
> > > Dear authors,
> > >
> > > I appreciate your detailed answers and your new additions to the paper. I will re-read everything carefully over the coming days and I will consider updating my score.

---

> ### Author Response · Authors · 2023-11-22
> **Additional response by the authors to the reviewer sCXL's response (1/2)**
>
> > For the calculation of the Fisher Information trace, do you use the same batch size as the one in training steps (i.e., 16, 16, 16, 256)?
>
> $\to$ Yes, you are correct.
>
> > I am wondering how accurate the Fisher Information trace estimate is if you use, for instance, a batch size of 16 once every four steps (as you write in your response).
>
>
> $\to$ Great question! To check how accurate our Fisher Information trace estimate is, we ran an experiment comparing the Fisher Information trace estimated with i) a batch size of 16 once every four steps and ii) a batch size of 64 for every step (*i.e.*, 16 times bigger sample size) on CIFAR-10 Gaussian task setup, and report the results in Sec.A.11 of the revision. From the result in Fig.5, we observe that the estimation with i) a batch size of 16 once every four steps does not deviate much from the estimation with ii) a batch size of 64 for every step, showing that our estimation is reasonably accurate.
>
> > Regarding my question about which computations are included in the computational budget of L-SAR, do you also include the cost of calculating the Fisher Information trace once every four training steps?
>
> $\to$ Yes, it includes the cost of computing the Fisher information. In particular, the computational cost is 2*(# parameters), *e.g.*, 0.92MFLOPs (=2 $\times$ 0.46M) in ResNet-32, since we calculate squared gradients for all parameters and sum them to calculate trace of fisher information per layer. Note that this cost is much smaller than forward and backward costs, which are 143.3MFLOPs and 286.6MFLOPs, respectively, for ResNet-32 for CIFAR-10 with a batch size of 16.
>
>
> > It seems to me that when you use Similarity-Aware Retrieval to sample a batch from the memory, you cannot use the same batch to compute the Fisher Information trace, since that batch is not sampled uniformly at random from the memory and the trace would be calculated incorrectly. What do you think?
>
> $\to$ We use the same batch because the actual training data distribution we use is sampled by similarity-aware retrieval rather than uniform. Since the Fisher Information is the amount of information the model parameters gain from training, it is natural to estimate it with the actual training distribution.

---

### Author Response · Authors · 2023-11-15
**General response**

We thank the reviewers for their helpful feedback and encouraging comments including supporting extensive experiments(**7DQF, htG8**), clear presentation(**7DQF**), well-motivated (**sCXL**), strong statistical theories (**htG8**).

We have uploaded the first revision of the manuscript (changes are highlighted by blue color).

---

### Author Response · Authors · 2023-11-22
**Second revision**

We have uploaded the second revision of the manuscript. The revision includes additional experiments with a new setup, to address the reviewers’ concerns and suggestions (sCXL).

Summary of the changes

- Add comparison of the Fisher Information trace estimated between i) a batch size of 16 once every four steps and ii) a batch size of 64 for every step on CIFAR-10 Gaussian task setup in Sec.A.11 of the revision (sCXL).

---

### Meta-Review · Area_Chair_ni7j · 2023-12-04

**Metareview:**

The manuscript was reviewed by three expert reviewers and they all unanimously recommended rejection. Authors responded to the reviews and reviewers interacted with these responses. After the discussion, decision remained as unanimous rejection. After careful consideration of the paper and the discussion, I also recommend rejection. The major issue behind the decision is presentation and clarity. The paper has significant issues with clarity and this limited the quality of the review process significantly. Original paper resulted in many misunderstandings. Although these issues are largely resolved, the manuscript still not in a publishable state. It needs a major re-write and an additional cycle of reviewing.

**Justification For Why Not Higher Score:**

The manuscript is not ready to be published or reviewed. It needs a thorough rewrote and a full review cycle.

**Justification For Why Not Lower Score:**

N/A

---

### Decision · Program_Chairs · 2024-01-16

Reject